# Effects of combining sensory-motor exercises with transcranial direct current stimulation on cortical processing and clinical symptoms in patients with lumbosacral radiculopathy: An exploratory randomized controlled trial

**Soheila Qanbari, Roya Khanmohammadi** 🔟 *, **Gholamreza Olyaei, Zohreh Hosseini, Hanie Sadat Hejazi**

Physical Therapy Department, Tehran University of Medical Sciences, Tehran, Iran

* rkhanmohammadi@sina.tums.ac.ir

## Abstract

### Background

Chronic low back pain (CLBP) is linked to reduced excitability in the primary motor (M1) and sensory (S1) cortices. Combining sensory-motor exercises with transcranial direct current stimulation (tDCS) to boost M1 and S1 excitability may improve treatment outcomes. This combined approach aligns with the neurophysiological mechanisms underlying CLBP and may target the neuroplastic changes induced by low back pain. This study aimed to assess whether enhancing M1 and S1 excitability via tDCS, alongside sensory-motor exercises, offers additional benefits for CLBP patients.

### Method

Participants were randomly assigned to receive either real or sham tDCS alongside sensory-motor exercises. Outcome measures included pain intensity, disability level, motor control ability, amplitudes of N80 and N150, and the amplitude of motor-evoked potential (MEP) and active motor threshold (AMT) for the multifidus (MF) and transversus abdominis/internal oblique (TrA/IO) muscles. A linear mixed-effects model (LMM) analyzed group, time, and interaction effects, while Spearman's correlation assessed relationships between neurophysiological and clinical outcomes.

### Results

The results showed significant reductions in pain intensity and disability levels (P < 0.001) and improved motor control (P < 0.001) in both groups. Both groups also exhibited increase in MF MEP amplitude (P = 0.042) and N150 amplitude (P = 0.028). The tDCS group demonstrated a significant decrease in AMT of MF and TrA/IO muscles (P < 0.05) and an increase in N80 amplitude (P = 0.027), with no significant changes in the control group. Additionally, the tDCS group had significantly lower AMT for the TrA/IO muscle in the post-test compared

**Data Availability Statement:** Data relevant to this study are available from Dryad at https://doi.org/10.5061/dryad.qz612jmr8.

**Funding:** This project was funded by the Tehran University of Medical Sciences (Grant No. 1401-2-103-55875).

**Competing interests:** The authors have declared that no competing interests exist.

to the sham group (P = 0.001). Increased N150 amplitude was correlated with improved motor control.

## Conclusions

The findings showed that sensory-motor exercises combined with either tDCS or sham tDCS effectively reduced pain intensity, decreased disability, and improved lumbar motor control in lumbosacral radiculopathy patients. No significant differences were observed between groups, indicating no added clinical benefit from tDCS over exercises alone. However, both groups demonstrated increased N150 and MF MEP amplitudes, suggesting enhanced cortical excitability in motor and sensory regions. While clinical outcomes were similar, neurophysiological data indicate that sensory-motor exercises play a central role in boosting cortical excitability, with tDCS further amplifying this effect, as evidenced by a significant AMT reduction in MF and TrA/IO muscles and an increase in N80 amplitude.

## 1. Introduction

Chronic low back pain (CLBP) is one of the most prevalent health issues worldwide, affecting up to 80% of individuals, with 5% to 10% suffering lumbosacral radiculopathy (LSR) [1]. This condition is characterized by radicular pain resulting from the compression or irritation of nerve roots in the lumbosacral region of the spine [2]. CLBP is a multifaceted condition influenced by mechanical, psychological, and psychosocial factors [3]. Consequently, many patients experience persistent radicular symptoms, with some requiring surgical intervention. However, not all patients achieve satisfactory pain relief, even with appropriate treatments [4]. This indicates that the underlying causes of persistent pain remain complex and not fully understood. Evidence suggests that treatments focused solely on correcting structural abnormalities often fail, highlighting the need to consider additional pathophysiological and biopsychosocial mechanisms [5].

Over the past decade, research has shown that the traditional structural-pathology paradigm, which assumes that dysfunctions originate locally at the injury site, is inadequate for understanding and treating chronic musculoskeletal disorders [6]. There is growing evidence that alterations in the central nervous system (CNS) play a crucial role in these disorders, emphasizing the importance of neurophysiological processes alongside structural and biomechanical factors in their development and progression [7–10].

Understanding the neurophysiological mechanisms associated with CLBP is essential for addressing the sensory-motor integration deficits linked to this condition. A critical component of this understanding involves the sensory cortex (S1) and the primary motor cortex (M1), which both show significant changes in individuals with CLBP. Studies have identified alterations in these cortical regions, including a decrease in gray matter density in S1 [8,11] and diminished activity in the secondary sensory cortex (S2) [9].

S1 is responsible for processing unimodal sensory signals and integrating them with motor signals for movement guidance, while S2 integrates multimodal sensory inputs bilaterally, which is essential for sensory-motor integration [12]. Notably, Jenkins et al. found that patients with lower S1 excitability during the acute phase of low back pain experienced more severe pain at a six-month follow-up than those with higher S1 excitability [13]. Another study

found that in CLBP patients, the S1 region is shifted, disrupting its connection with M1 and possibly impairing postural control [14].

In light of these findings, the M1 also emerges as a critical area of focus. Studies by Strutton PH et al. and Tsao H et al. have shown that patients with CLBP exhibit reduced M1 excitability and reduced map volume of multifidus muscle compared to healthy individuals [10,15]. This reduced excitability correlates with higher pain levels, slower recovery [13] and increased functional disability [10]. Additionally, Li et al. found that the cortical representation of the transversus abdominis (TrA) and multifidus (MF) muscles in the left motor cortex is more discrete in CLBP patients, leading to reduced muscle coordination and lumbar stability [16]. A recent study by Masse-Alarie et al. reported disrupted sensorimotor integration in CLBP individuals, suggesting that sensory input from the lower back enhances corticospinal excitability more than in pain-free individuals. They recommended interventions targeting both sensory processing and spinal motor control [17]. These interventions could include exercises that promote top-down influences on sensorimotor integration or brain stimulation techniques that directly target these pathways.

These findings suggest that alterations in sensorimotor integration, along with decreased excitability in S1 and M1, may contribute to chronic and recurrent low back pain. This maladaptive plasticity in the sensory and motor cortices helps explain the limited effectiveness of conventional treatments for these patients. Most of the existing research has focused on non-specific CLBP, while investigations into changes in M1 and S1 among LSR patients are still limited. However, studies indicate that brain networks are also affected in these patients [4,18].

Effective sensory-motor control of the spine relies on precise central processing and integration of sensory information into motor commands. Thus, maladaptive plasticity in the sensory and motor cortices can disrupt these functions, thereby impairing spinal control [6,12]. In other words, alterations in sensory-motor processing within the CNS may underlie compromised sensory-motor control of the back, potentially contributing to persistent pain and disability [12].

From a therapeutic perspective, improving sensory-motor control in CLBP patients involves therapeutic exercises as a frontline treatment [19]. Recent studies emphasize the importance of sensory-motor exercises that address both sensation and movement simultaneously [20,21]. These exercises are designed to normalize sensory and motor alterations, thereby alleviating pain [21]. Sensory-motor exercises enhance proprioceptive inputs, refine motor responses to unexpected stimuli, improve postural control, and alleviate musculoskeletal pain [21–23]. Some studies have demonstrated that such exercises can enhance brain activity and muscle control through the development of neuroplasticity in the cerebral cortex, leading to pain reduction in individuals with CLBP [24–26].

Given that some patients with LBP exhibit altered organization in the M1 and S1 areas, combining sensory-motor exercises with adjunct therapies that directly boost M1 and S1 excitability could further improve treatment outcomes. This combined approach appears to be consistent with the underlying neurophysiological mechanisms in CLBP and may effectively target the neuroplastic changes induced by pain. Transcranial direct current stimulation (tDCS) is a non-invasive technique that modulates cortical excitability depending on the polarity of the stimulation [27]. Anodal tDCS generally enhances cortical excitability, and multi-session tDCS can modulate long-term potentiation or long-term depression, both essential for neural plasticity [3]. Thus, combining sensory-motor exercises with tDCS could effectively integrate both central and peripheral mechanisms.

Evidence suggests that brain stimulation targeting the motor or sensory cortices can alleviate pain in patients with CLPB [3,28–32]. Additionally, a study by Attal et al. on patients with LSR found that repetitive transcranial magnetic stimulation (rTMS) over M1 was more

effective than tDCS and sham treatments, potentially modulating both the sensory and affective aspects of pain [33]. Two studies have also investigated the effects of trainings combined with tDCS in CLBP patients [34,35]. These studies showed that adding M1 anodal tDCS to training significantly improved pain, postural stability, and psychological well-being.

While existing studies provide valuable insights, they primarily focus on stimulating the M1 and overlook the fact that individuals with CLBP also exhibit reduced excitability in the S1. Given the alterations in both regions and S1's crucial role in sensory integration and pain processing, targeting both M1 and S1 may offer a more comprehensive and effective strategy for alleviating CLBP symptoms.

Despite the recognition of sensory-motor integration deficits as a significant factor in CLBP, research on interventions targeting both sensory and motor cortices remains limited. Combining tDCS of these areas could enhance the efficacy of sensory-motor exercises by improving excitability in both domains, potentially leading to more substantial relief in CLBP management.

Therefore, this study aims to address this gap by investigating whether enhancing excitability in M1 and S1 through anodal tDCS, in conjunction with sensory-motor exercises, can improve sensory-motor brain processing. This approach may subsequently alleviate pain, reduce disability, and enhance lumbar motor control in patients with CLBP. To our knowledge, no previous studies have explored the combined effects of M1 and S1 stimulation with sensory-motor exercises on clinical symptoms in individuals with LSR.

Additionally, this study employed neurostimulation techniques to measure the excitability of both the sensory and motor cortices. Another aim was to investigate the relationship between neurophysiological parameters and clinical outcomes. We hypothesized that while both groups undergoing sensory-motor exercises would experience beneficial effects, the effectiveness would be further enhanced with the addition of tDCS. Specifically, it was expected that pain and disability would decrease, motor control ability would improve, sensory and motor-evoked potential amplitudes would increase, and active motor thresholds would decrease, indicating increased excitability of the sensory and motor cortex.

## 2. Materials and methods

### 2.1. Study design

This study was a randomized, single-blinded, and controlled trial. Participants were randomly assigned to receive either real or sham tDCS alongside sensory-motor exercises. The outcome measures were taken at pre and post-treatment. Post-treatment evaluations were carried out within 24 to 48 hours following the completion of the intervention. The independent variables included group allocation (real tDCS versus sham tDCS) and time (pre-treatment and post-treatment). Primary outcome measures comprised pain intensity, disability level, and lumbar motor control ability. Additionally, secondary outcomes encompassed the amplitude of N80 and N150 (sensory-evoked potential, SEP), as well as the amplitude of motor-evoked potential (MEP) and active motor threshold (AMT) for multifidus and transversus abdominis/internal oblique muscles to measure sensory and motor cortex excitability. Both groups received the treatment in a laboratory setting. This study was registered as a clinical trial on the Iranian Registry of Clinical Trials (IRCT20211222053484N1) on February 16, 2022.

### 2.2. Participants

Participants with chronic unilateral LSR secondary to L4/L5 and L5/S1 disc herniation (confirmed on magnetic resonance imaging) were included in the study (Table 1). A flow diagram based on the CONSORT statement shows participants from enrollment to analysis (Fig 1). All

**Table 1. The demographical and clinical characteristics of subjects at baseline.**

| Outcome Measures | Intervention (N = 17) | | Control (N = 17) | | P Value |
|---|---|---|---|---|---|
| | Mean / Median | SD / Q1–Q3 | Mean / Median | SD / Q1–Q3 | |
| **Female (N (%))✦** | 6 (35) | | 10 (59) | | 0.17 |
| **Right Side of Pain (N (%))✦** | 6 (35) | | 10 (59) | | 0.17 |
| **Age (years)″** | 35.71 | 8.53 | 42.53 | 11.41 | 0.06 |
| **Height (cm)\*** | 170.00 | 158.50–182.50 | 160.00 | 157.50–169.00 | 0.19 |
| **Weight (kg)\*** | 70.00 | 61.50–95.50 | 74.00 | 61.50–84.00 | 0.76 |
| **BMI (kg/m2)″** | 25.05 | 3.83 | 25.59 | 4.96 | 0.72 |
| **Pain Intensity (0–10)″** | 6.41 | 1.91 | 6.29 | 1.90 | 0.86 |
| **Disability Level (0–50)″** | 12.47 | 6.62 | 14.47 | 9.54 | 0.48 |
| **Pain Duration (Months)\*** | 12.00 | 7.00–42.00 | 36.00 | 12.00–90.00 | 0.17 |

Pain intensity and disability level were measured by visual analog scale and Oswestry low back pain disability questionnaire, respectively.

Q1 and Q3 represent the first and third quartiles of a parameter.

✦ Chi-square test was used.

″ Independent t-test was used.

\* Wilcoxon rank-sum test was applied.

participants provided written informed consent. This study was approved by the Institutional Ethics Committee of the Tehran University of Medical Sciences (IR.TUMS.FNM. REC.1400.170). The subjects were recruited between March 6, 2022, and August 15, 2023.

The inclusion criteria were (1) between the ages of 20 and 60 years; (2) unilateral radicular pain lasting for more than three months or recurrent pain with at least three episodes of more

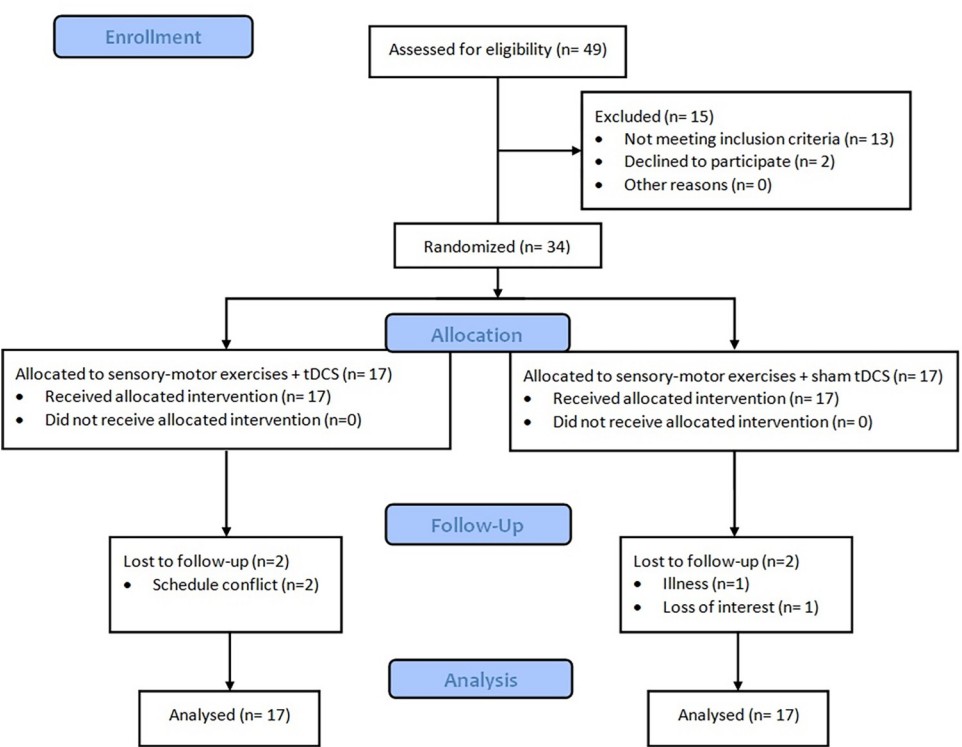

**Fig 1. The flow of the participants from enrollment to analysis.**

than one week in the past year [36]; (3) leg pain greater than back pain; (4) having pain in the distribution of the sciatic nerve (the anterolateral aspect of the leg and the dorsum of the foot or the posterior aspect of the leg extending to the heel and lateral aspect of the foot that indicate L4/L5 and L5/S1 root compression, respectively [37]; (5) pain radiation in clinical tests such as slump, straight leg raising, or Lasègue's sign [37]; (6) L4/L5 or L5/S1 disc herniation in magnetic resonance imaging; (7) having a pain score > 3 out of 10 on a visual analog scale (VAS) in the week prior to enrolment (moderate to severe pain); (8) no self-reported history of spinal fracture, and surgery; (9) no psychological or neurological disorders; (10) no history of migraines; (11) no history of epilepsy or seizure; (12) no history of head injury resulting in a loss of consciousness; (13) not having metallic implants, including surgical clips, intracranial electrodes, or a pacemaker; and (14) not being on prescription medication, or self-medicating such as anxiolytic, antidepressant, or analgesic medications which can affect neuronal excitability [38]. Additionally, patients with severe nerve root compression that had bowel or bladder dysfunction, progressive neurological deficits, or cauda equine syndrome were not included, as it was believed that these participants might benefit more from surgery.

## 2.3. Sample size

The sample size was determined using G*Power 3.1.9.2 software, following the methodology of the Jafarzadeh et al. study [34], with pain as the primary outcome measure. The calculation utilized a between-group Cohen's $d$ of 1.09, derived from the mean difference of 1.42 divided by the pooled standard deviation of 1.30, with a power of 0.8 and an alpha level of 0.05. This power analysis indicated that a minimum of 30 participants was required. To accommodate a potential dropout rate of 15%, a total of 34 participants were recruited.

## 2.4. Randomization and blinding

Participants were randomly assigned to either the intervention group (sensory-motor exercises plus real tDCS) or the control group (sensory-motor exercises plus sham tDCS) with a 1:1 allocation ratio. Randomization was conducted using a web-based randomization service (www. randomization.com) with block randomization (block size = 4) managed by an independent third party. The allocation details were written on cards and placed in sequentially numbered, opaque, sealed envelopes to conceal the sequence from the primary researchers. Participants in this study were blinded to their group assignment.

## 2.5. Assessments

**2.5.1. Pain intensity.** To evaluate pain intensity, the Visual Analog Scale (VAS) was used. This scale uses a 10 cm line on which participants rate their mean pain over the past week, with zero representing no pain and ten representing the most unbearable pain [39].

**2.5.2. Disability level.** The Oswestry Disability Index questionnaire was used to assess disability level. This questionnaire evaluates functional disability in individuals with back pain and includes ten sections covering aspects such as pain intensity, personal care, social life, sex life, walking, lifting, standing, sitting, sleeping, and travelling. Each section is scored from 0 to 5, with the total score calculated by summing the individual scores. A score of 0–4 indicates no disability, 5–14 indicates mild disability, 15–24 indicates moderate disability, 25–34 indicates severe disability, and 35–50 indicates complete disability [40].

**2.5.3. Lumbar motor control ability.** A test battery consisting of six assessments developed by Luomajoki was utilized to evaluate lumbar motion control ability [41]. These tests are reliable for diagnosing motor control dysfunctions (kappa value > 0.6) [42]. Participants received instructions on how to perform each movement and were then asked to execute

them. If a movement was performed incorrectly or if the participant did not understand the instructions, the examiner provided further explanation and demonstration. Participants were allowed three attempts per movement. If a participant could correct the movement with guidance, no motor control dysfunction was recorded (score 0). However, if the participant could not perform the movement correctly despite the guidance, the test was considered positive (score 1). Thus, a score of 0 indicates all tests were performed correctly, while a score of 6 indicates none were performed correctly [41].

**2.5.4. Motor cortex excitability.** *2.5.4.1. Surface electromyography (EMG).* Surface EMG activity of the multifidus (MF) and transversus abdominis/internal oblique (TrA/IO) muscles was recorded using disposable, pre-gelled Ag/AgCl snap electrodes (Noraxon USA Inc, Arizona, USA). For the MF muscle, electrode was placed at the level of the L5 spinous process, along the line connecting the posterior superior iliac spine and the L1-L2 space on the painful side [43]. For the TrA/IO muscle, electrode was placed 2 cm below and 2 cm inside the anterior superior iliac spine on the painful side, targeting the most superficial part of these muscles [44]. The ground electrode was placed on the iliac crest. EMG signals were sampled at 2000 Hz, amplified by 1000x, filtered (5–2000 Hz) and saved for further analysis (Motion Lab Systems, MA400-22).

*2.5.4.2. Transcranial magnetic stimulation (TMS).* A 95 mm outer diameter D-B80 Butterfly Coil (Tonica Elektronik A/S, Denmark) connected to a MagPro X100 stimulator with MagOption (MagVenture, Farum, Denmark) was used to measure motor cortex excitability.

To record the variables related to the MF muscle, the patient sat comfortably in a chair with feet flat on the floor. To achieve maximal MF muscle contraction, the participant was instructed to cross the arms over the chest and perform trunk extension against manual resistance, maintaining this posture for 3 seconds [45]. This movement was repeated three times, and the highest root mean square (RMS) within a 1-second window was considered as the maximal contraction. Subsequently, 20% of this value was set as the sub-maximal contraction. [45]. This target contraction level was displayed on a monitor to ensure that the participant maintained the specified contraction level throughout the recording.

For the TrA/IO muscle, participants sat with arms resting on the chair and knees extended. Maximum contraction was achieved by performing a forced exhalation maneuver, maintained for 3 seconds and repeated three times. Subsequently, 15% of this maximum value was set as the sub-maximal contraction level which needed to be sustained throughout the recording [44].

Stimulation was applied on the side opposite to the painful region. The Cz (vertex) was determined using the international 10/20 system, and the coil center was placed 2 cm lateral to Cz at a 45-degree angle to the anterior-posterior axis to direct the current from back to front [44]. Participants wore a tight-fitting bathing cap, and Cz and the hotspot were marked to ensure consistent stimulation sites during pre- and post-treatment sessions.

The active motor threshold (AMT) was determined as the lowest TMS intensity evoking a motor potential with a minimum amplitude of 100 μV in at least 5 out of 10 repetitions [44]. Starting intensity was set at 30% of the stimulator output and adjusted in increments of 1–3% until achieving the required MEP amplitude ($\geq$100 μV). The stimulation intensity was then set to 120% of the AMT, and the peak-to-peak amplitude of the resulting biphasic waveform was recorded as the motor-evoked potential amplitude [43]. For MEP amplitude, we administered 10 stimulations at 120% of AMT to the hotspot and then calculated the mean of the resulting responses from all 10 stimulations.

**2.5.5. Sensory cortex excitability.** To investigate sensory cortex excitability, sensory-evoked potentials were measured using the EMG/NCV/EP 5000Q device (NR Sign Inc, IR). This experiment focused on two outcome measures: amplitudes N80 and N150. The N80

amplitude is believed to represent processing in the S1, while the N150 amplitude is associated with processing in the S2 [13].

For the experiment, participants sat comfortably in a chair with feet flat on the floor and hands in a relaxed position. They were asked to keep the eyes closed but remain awake. Gold-plated cup electrodes were placed on the scalp to record SEPs from the S1 region, on the side opposite the participant's pain site. According to the international 10–20 system, this location is 3 cm lateral and 2 cm posterior to Cz. A reference electrode was placed at the Fz region, and a ground electrode was positioned on the forehead [13,46]. The electrode impedance was kept below 5 k Ohm.

Electrical stimulation was applied using a bipolar electrode on the lower back, 3 cm lateral to the L3 spinous process on the painful side, with a pulse duration of 1 millisecond and a frequency of 2 Hz. The bandwidth was set from 1 to 500 Hz. The stimulation intensity started at 1 mA and was increased by 1 mA increments until reaching the perception threshold. The intensity was then set to three times the perception threshold. If this intensity caused pain, it was reduced by 1 mA increments until it was no longer painful [13,46]. Stimuli were applied 500 times, and this process was repeated twice. The average of these two repetitions was used for the final analysis. The N80 amplitude was defined as the largest peak within 40 to 90 milliseconds after stimulation, while the N150 amplitude was the largest peak within 90 to 180 milliseconds [47]. In this study, N80 and N150 were typically observed around 65 and 120 milliseconds, respectively (Fig 2).

## 2.6. Groups

The groups were (1) the intervention group (sensory-motor exercises plus real tDCS) and (2) the control group (sensory-motor exercises plus sham tDCS). Participants completed 12 sessions over four weeks, with three sessions per week, each lasting 60 minutes. All exercises were done individually and under the supervision of a therapist.

**2.6.1. Sensory-motor exercises.**   In sensory-motor exercises, the goal was to stimulate the mechanoreceptors in three areas (the sole of the feet, sacroiliac joints, and cervical spine) to enhance afferent input to the sensory-motor system. Therefore, it was essential to maintain the correct position of these three areas throughout all exercises. To stimulate plantar receptors, the exercises were performed barefoot. Participants were instructed to contract the foot intrinsic muscles to increase the medial longitudinal arch of the foot without flexing the toes. Throughout the exercises, it was important to ensure that the sacroiliac joints and cervical spine remained in a neutral alignment. Additionally, participants were asked to gently draw the umbilicus inward and tuck the chin to activate the transversus abdominis and the deep neck flexor muscles, respectively. Overall, the exercises consisted of two phases: static and dynamic [23,48,49].

Static phase: In this phase, the emphasis was placed on stabilizing the pelvis through the activation of the diaphragm, MF, pelvic floor, and TrA muscles. This served as a foundation for the subsequent movements of the upper and lower limbs, aligning with the principle of "proximal stability for distal mobility." Progression in this phase involved transitioning the base of support (BOS) from bilateral to unilateral stance, and then to a "half-step" position. The half-step was a position in which the participant brought the trunk forward over the short foot while maintaining neutral alignment of the cervical and lumbar spine. Furthermore, BOS advancement included transitioning from a firm surface to a foam surface, and then to rocker and wobble boards. Additionally, the center of gravity (COG) was challenged through weight shifts and perturbations induced manually or with elastic resistance bands. These scenarios elicited automatic and reflexive postural reactions [23,48,49].

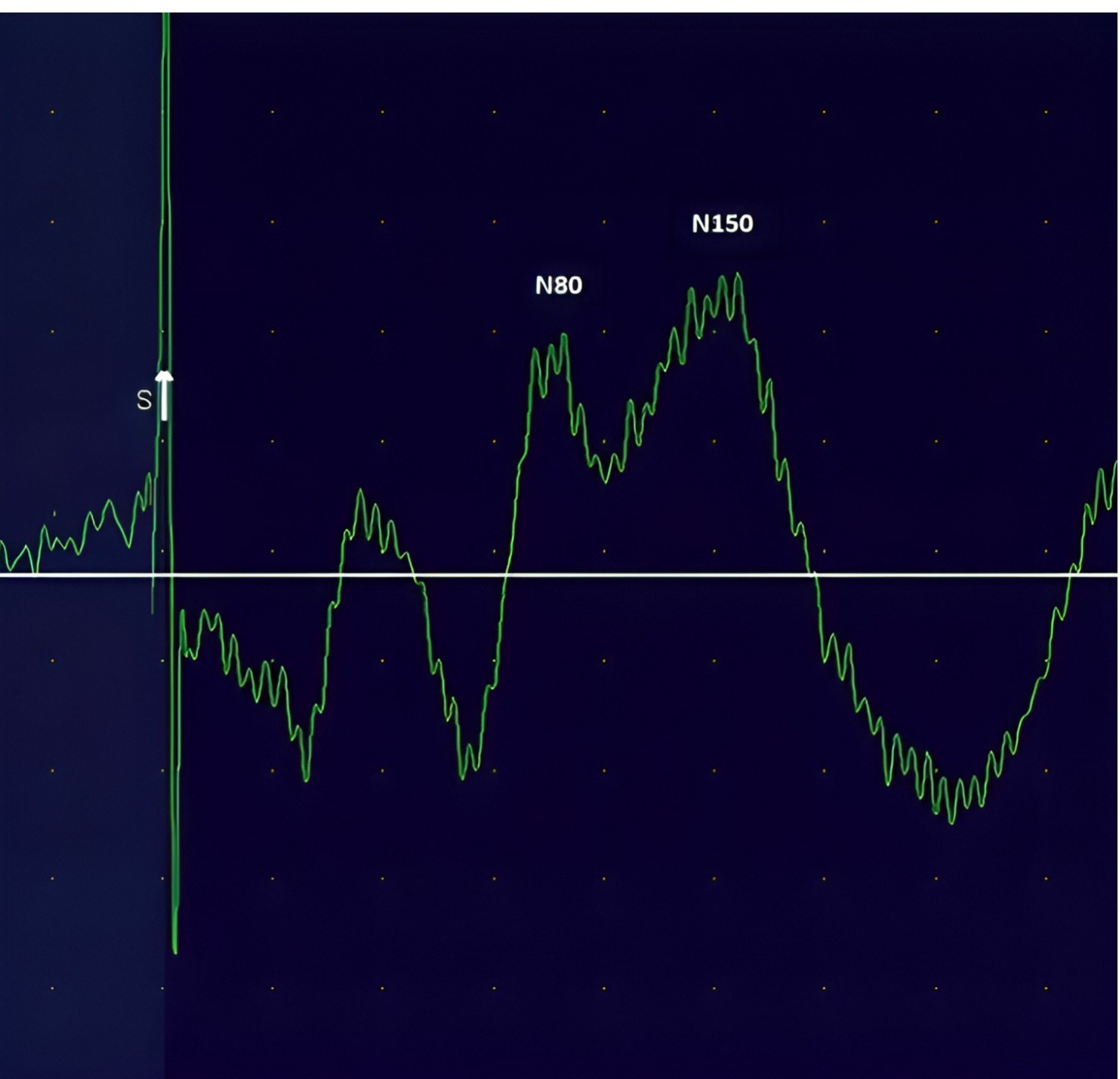

**Fig 2. Sensory-evoked potentials of N80 and N150.**

Dynamic phase: This phase commenced once the participant could consistently maintain pelvic stability in the half-step position under varying conditions. Similar to the static phase, the progression of the BOS was continued, while the COG was further challenged by incorporating movements of the upper and lower limbs, such as throwing and shooting balls, or by gradually increasing resistance. These activities triggered feed-forward mechanisms. In sensory-motor exercises, emphasis was placed on the quality rather than the quantity of exercises. As a result, participants could advance to a higher level once they demonstrated proficiency at the current level, regardless of the number of repetitions completed. In essence, the criterion

for progression was not a specific number of repetitions, but rather the quality of the exercises [23,48].

**2.6.2. Transcranial direct current stimulation (tDCS).** Participants received stimulation using NEUROSTIM2, utilizing two anode electrodes (2*4) and two cathode electrodes (4*4) saturated with sterile saline (0.9% NaCl). One anode electrode was placed over M1 (C3/C4) according to the international 10–20 system. Another anode electrode was positioned over S1 (2 cm posterior to C3/C4). The cathode electrodes served as reference electrodes and were positioned over the contralateral supraorbital area (Fig 3) [49,50]. The anode electrodes were

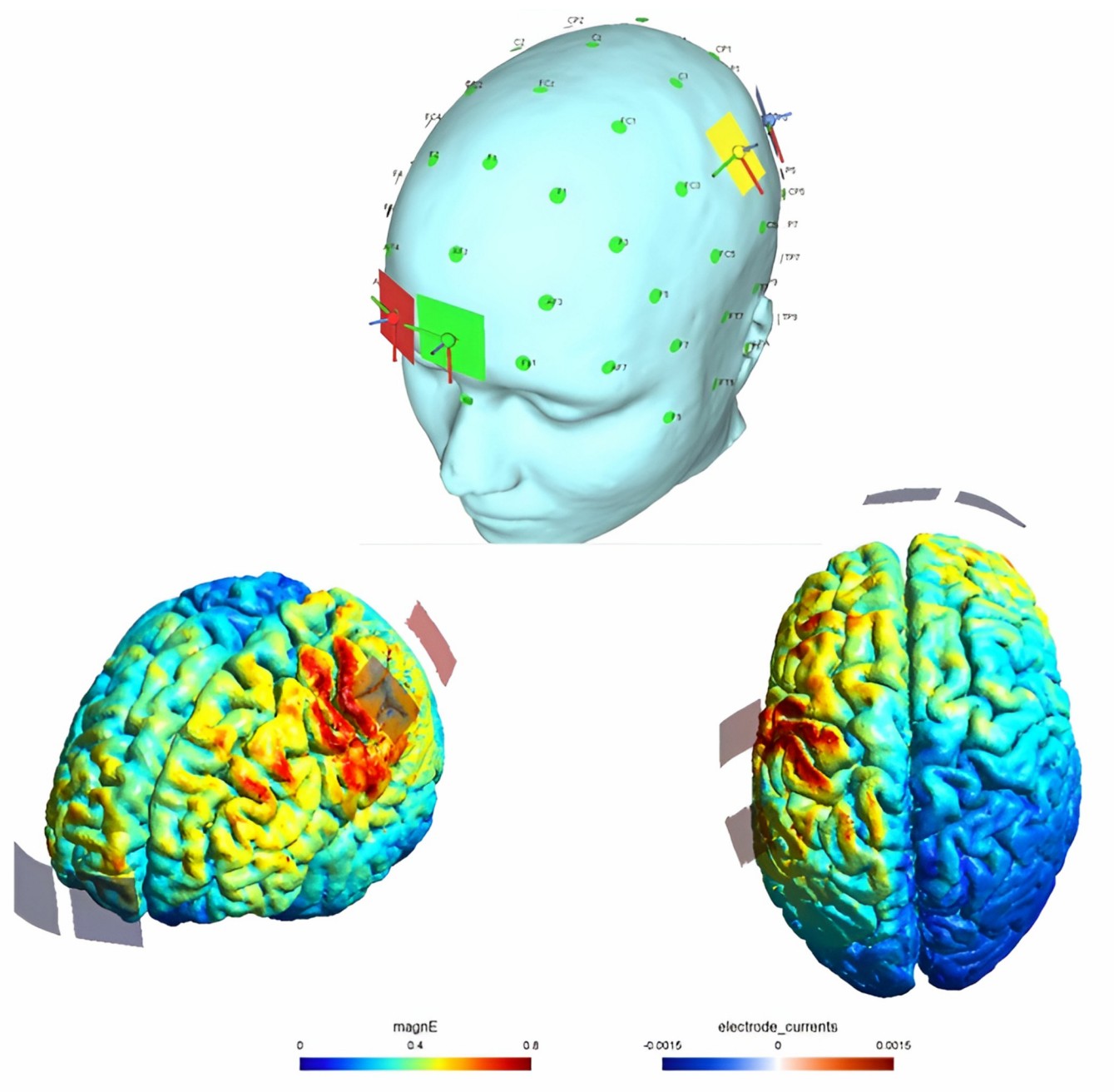

**Fig 3. Electrode montage and electric field distribution simulated with SimNIBS software.**

situated contralateral to the side experiencing pain, while the cathode electrodes were placed ipsilateral to the pain side. The current intensity was set to 1.5 mA and delivered for 20 minutes, resulting in a current density at the active electrodes of 0.188 mA/cm2. This current density falls below the safety limit reported by Bikson et al. for human subjects (25.46 A/m2) [51]. To minimize discomfort, the current was gradually ramped up (fade-in) or down (fade-out) during the first and last 10 seconds, respectively. In sham stimulation, the electrode placement was identical to real stimulation, but after a 10-second fade-in period, no current was applied for the duration of 20 minutes.

It's important to mention that, according to the 2016 evidence-based update, the application of tDCS in human trials (up to 40 minutes, up to 4 mA, up to 7.2 Coulombs) has not been associated with any significant adverse effects or permanent harm [52].

## 2.7. Statistical analysis

For statistical analysis, IBM SPSS Statistics version 23.0 was utilized. The Shapiro-Wilk test was used to evaluate the normality of the data distribution. Most data followed a normal distribution, except for certain demographic characteristics like height, weight, and pain duration. The Chi-square test was used to compare the proportions of gender and pain side between the groups. For parameters with a normal distribution, an independent t-test was conducted, while the Wilcoxon rank-sum test was applied to non-normally distributed parameters. The results indicated no significant baseline differences between the two groups, confirming their comparability before the intervention.

To examine the main effects of group (sensory-motor exercises + tDCS vs. sensory-motor exercises + sham tDCS), time (pre- and post-treatment), and their interactions, a linear mixed effects model (LMM) was performed. When significant interactions were identified, the groups were analyzed separately for more detailed insights. Time, group, and their interaction effects were considered fixed effects, while participants were treated as random effects. Mixed models are generally favored over traditional general linear models for handling missing data, as they can utilize all data points, even when some observations are absent. This model evaluates changes over time using maximum likelihood estimation, thus applying an intention-to-treat approach that yields data more reflective of real-world conditions. Furthermore, Cohen's *d* effect sizes were interpreted as small = 0.2–0.5, moderate = 0.5–0.8, large = 0.80–1.3, and very large ≥ 1.3 [53].

Additionally, to examine the relationship between treatment-induced changes (post-test minus pre-test) in neurophysiological and clinical outcomes, Spearman's correlation coefficient was employed, as the change values did not follow a normal distribution. These correlations were computed for all participants regardless of their group assignment. Given that the analysis involved 3 clinical parameters (pain intensity, disability level, and lumbar motor control ability) and 6 neurophysiological parameters (N80 and N150 amplitudes, MEP amplitude, and AMT for multifidus and transversus abdominis/internal oblique muscles), a total of 18 correlations (3×6) were tested. To account for the risk of Type I errors from multiple comparisons, the Bonferroni correction was applied, resulting in a corrected p-value threshold of 0.003. The correlation strength was assessed using the correlation coefficient r, which was categorized as follows: a value between 0.80 and 1.0 indicates a very strong correlation; 0.60 to 0.79 signifies a strong correlation; 0.40 to 0.59 represents a moderate correlation; 0.20 to 0.39 suggests a weak correlation; and values less than 0.20 reflect a negligible correlation [54].

## 3. Results

The descriptive data and the results of LMM are presented in Tables 2 and 3, respectively.

**Table 2. The mean and standard deviation of outcome measures.**

| Outcome Measures | Intervention | | | | | Control | | | | |
|---|---|---|---|---|---|---|---|---|---|---|
| | Pre-Test | | Post-Test | | MD (95% CI) | Pre-Test | | Post-Test | | MD (95% CI) |
| | Mean | SD | Mean | SD | | Mean | SD | Mean | SD | |
| **Primary Outcome Measures** | | | | | | | | | | |
| **Pain Intensity (0–10)** | 6.41 | 1.91 | 2.53 | 1.77 | 3.87 (2.72, 5.01) | 6.29 | 1.90 | 2.87 | 1.64 | 3.73 (2.15, 5.32) |
| **Disability Level (0–50)** | 12.47 | 6.62 | 7.40 | 4.27 | 5.33 (2.54, 8.13) | 14.47 | 9.54 | 5.20 | 3.23 | 10.07 (4.04, 16.09) |
| **Motor Control Test (0–6)** | 3.59 | 1.58 | 2.07 | 0.88 | 1.73 (1.16, 2.31) | 3.82 | 1.07 | 2.20 | 1.32 | 1.60 (1.05, 2.15) |
| **Secondary Outcome Measures** | | | | | | | | | | |
| **MEP Amplitude of MF (µV)** | 168.12 | 22.89 | 197.10 | 43.14 | -29.43 (-50.58, -8.29) | 169.71 | 57.79 | 174.73 | 26.49 | -12.80 (-42.13, 16.53) |
| **MEP Amplitude of TrA/IO (µV)** | 200.32 | 59.70 | 218.71 | 81.33 | -19.74 (-37.68, -1.80) | 189.41 | 59.14 | 201.33 | 94.17 | -10.07 (-47.24, 27.11) |
| **AMT of MF (%MOS)** | 54.47 | 9.91 | 48.53 | 6.79 | 5.00 (1.70, 8.30) | 54.18 | 10.44 | 54.67 | 11.25 | -0.87 (-4.22, 2.49) |
| **AMT of TrA/IO (%MOS)** | 48.24 | 8.83 | 43.47 | 5.91 | 4.87 (-0.55, 10.28) | 52.18 | 8.95 | 53.80 | 9.78 | -1.60 (-5.03, 1.83) |
| **N80 Amplitude (µV)** | 2.06 | 0.70 | 2.76 | 1.63 | -0.74 (-1.43, -0.06) | 2.06 | 0.90 | 2.11 | 1.05 | 0.01 (-0.37, 0.38) |
| **N150 Amplitude (µV)** | 1.35 | 0.66 | 1.77 | 1.31 | -0.46 (-1.06, 0.14) | 1.22 | 0.53 | 1.43 | 0.81 | -0.19 (-0.39, 0.02) |

MEP: Motor-Evoked Potential; AMT: Active Motor Threshold; MF: Multifidus; TrA/IO: transversus abdominus/internal oblique; MOS: Maximum Stimulator Output; MD: Mean Differences; CI: Confidence Interval.

## 3.1. Clinical data

**3.1.1. Pain intensity.** The interaction effect and group effect were not significant, indicating that the change in pain intensity over time was similar for both groups, with neither group showing superiority. However, the time effect was significant, meaning that pain intensity

**Table 3. The results of linear mixed model.**

| Outcome Measures | Time Effect | | | Group Effect | | | Interaction Effect | | |
|---|---|---|---|---|---|---|---|---|---|
| | F | P | Cohen's d | F | P | Cohen's d | F | P | Cohen's d |
| **Primary Outcome Measures** | | | | | | | | | |
| Pain Intensity | 71.522 | < **0.001**\* | 1.980 | 0.053 | 0.820 | 0.126 | 0.259 | 0.614 | 0.179 |
| Disability Level | 25.335 | < **0.001**\* | 1.595 | 0.007 | 0.932 | 0.251 | 2.214 | 0.146 | 0.515 |
| Motor Control Test | 85.500 | < **0.001**\* | 2.316 | 0.467 | 0.499 | 0.194 | 0.026 | 0.873 | 0.058 |
| **Secondary Outcome Measures** | | | | | | | | | |
| MEP Amplitude of MF | 4.568 | **0.042**\* | 0.230 | 0.730 | 0.400 | 0.367 | 1.680 | 0.206 | 0.500 |
| MEP Amplitude of TrA/IO | 2.656 | 0.113 | 0.290 | 0.446 | 0.509 | 0.231 | 0.233 | 0.633 | 0.172 |
| AMT of MF | 4.295 | **0.047**\* | 0.193 | 0.736 | 0.397 | 0.519 | 7.992 | **0.008**\* | 1.030 |
| AMT of TrA/IO | 1.296 | 0.263 | 0.283 | 8.135 | **0.007**\* | 0.954 | 5.202 | **0.029**\* | 0.801 |
| N80 Amplitude | 4.714 | **0.037**\* | 0.023 | 1.165 | 0.288 | 0.517 | 4.319 | **0.046**\* | 0.729 |
| N150 Amplitude | 5.279 | **0.028**\* | 0.347 | 0.970 | 0.331 | 0.344 | 0.802 | 0.377 | 0.313 |

MEP: Motor-Evoked Potential; AMT: Active Motor Threshold; MF: Multifidus; TrA/IO: transversus abdominus/internal oblique; MOS: Maximum Stimulator Output.

Cohen's $d$ effect sizes were interpreted as small = 0.2–0.5, moderate = 0.5–0.8, large = 0.80–1.3, and very large $\geq$ 1.3.

\* Significant differences (P < 0.05).

decreased significantly from pre-test to post-test in both groups, $F(1,32.224) = 71.522$, $P < 0.001$, Cohen's $d = 1.98$.

**3.1.2. Disability level.** The interaction effect and group effect were not significant, indicating that the change in disability level over time was similar between the two groups, with neither group being superior. However, the time effect was significant, meaning that the disability level significantly decreased from pre-test to post-test in both groups, $F(1,33.416) = 25.335$, $P < 0.001$, Cohen's $d = 1.595$.

**3.1.3. Lumbar motor control ability.** The interaction and group effects were not significant, but the time effect was significant. LMM demonstrated a significant improvement in lumbar motor control ability (as shown by a decrease in Luomajoki test scores) from pre-test to post-test in both groups, $F(1,30.750) = 85.5$, $P < 0.001$, Cohen's $d = 2.316$.

## 3.2. Neurophysiological data

**3.2.1. Motor cortex excitability.** For MEP amplitude of MF, the interaction and group effects were not significant, whereas time effect was significant. LMM showed that the MEP amplitude of MF significantly increased from pre-test to post-test in both groups, $F(1,26.793) = 4.568$, $P = 0.042$, Cohen's $d = 0.23$.

For the AMT of MF, the analysis revealed a significant interaction effect, $F(1, 30.367) = 7.992$, $P = 0.008$, Cohen's $d = 1.03$, although the main effect of group was not significant. Separate analyses for each group indicated that, in the sensory-motor exercise combined with tDCS group, there was a significant time effect, $F(1, 14.955) = 12.325$, $P = 0.003$, Cohen's $d = 1.81$, showing a significant reduction in AMT from pre-test to post-test. In contrast, the control group exhibited no significant change over time, $F(1, 15.274) = 0.289$, $P = 0.599$, Cohen's $d = 0.274$.

For the AMT of TrA/IO, both the interaction and group effects were significant. After separating the groups and analyzing them individually, it was found that in the intervention group, the AMT significantly decreased from pre-test to post-test, $F(1,16.499) = 4.094$, $P = 0.049$, Cohen's $d = 0.997$. However, there was no significant change over time in the control group, $F(1,15.765) = 1.108$, $P = 0.308$, Cohen's $d = 0.530$. Moreover, sensory-motor exercises combined with tDCS demonstrated a significantly lower AMT in the post-test compared to those combined with sham tDCS, $F(1,30) = 13.142$, $P = 0.001$, Cohen's $d = 1.324$.

**3.2.2. Sensory cortex excitability.** Regarding N80 amplitude, the interaction effect was significant, indicating that the change in N80 amplitude over time differed between the two groups. However, main effect of group was not significant. After separating the groups and analyzing them individually, it was found that in the intervention group, N80 amplitude significantly increased from pre-test to post-test, $F(1,16.513) = 5.860$, $P = 0.027$, Cohen's $d = 1.191$. However, this change was not significant in the control group, $F(1,15.743) = 0.002$, $P = 0.967$, Cohen's $d = 0.021$.

For N150 amplitude, the LMM indicated that the interaction and group effects were not significant, while the time effect was significant. Specifically, the N150 amplitude showed a significant increase from pre-test to post-test in both groups, $F(1,32.687) = 5.279$, $P = 0.028$, Cohen's $d = 0.347$.

## 3.3. Linear relationships

The findings revealed a significant inverse correlation between changes in lumbar motor control and changes in N150 amplitude post-treatment, $P = 0.002$, $r = -0.549$. Indeed, higher N150 amplitude after treatment was associated with lower scores on the motor control test

**Table 4. Linear relationships (Spearman's r (p-value)) the treatment-induced changes in clinical and neurophysiological data.**

| Parameters | Pain Intensity | Disability Level | Motor Control Test |
|---|---|---|---|
| MEP Amplitude of MF | -0.266 (0.156) | -0.105 (0.579) | -0.174 (0.358) |
| MEP Amplitude of TrA/IO | -0.152 (0.424) | -0.209 (0.267) | -0.187 (0.323) |
| AMT of MF | 0.276 (0.139) | -0.045 (0.813) | 0.200 (0.290) |
| AMT of TrA/IO | -0.137 (0.469) | 0.027 (0.887) | -0.057 (0.764) |
| N80 Amplitude | -0.127 (0.504) | 0.050 (0.794) | 0.011 (0.955) |
| N150 Amplitude | -0.208 (0.269) | -0.049 (0.796) | **-0.549 (0.002)**\* |

MEP: Motor-Evoked Potential; AMT: Active Motor Threshold; MF: Multifidus; TrA/IO: transversus abdominus/internal oblique.

r is effect size (Very strong = 0.80–1.0; strong = 0.60–0.79; moderate = 0.40–0.59; weak = 0.20–0.39; negligible <0.20).

\* Significant differences (P < 0.003).

(indicating better performance). In simpler terms, an increase in amplitude correlated with improved motor control (Table 4 and Fig 4).

However, there were no significant correlations between the other clinical and neurophysiological parameters, with correlation strengths ranging from 0.011 to 0.276 and corresponding p-values between 0.139 and 0.955 (Table 4).

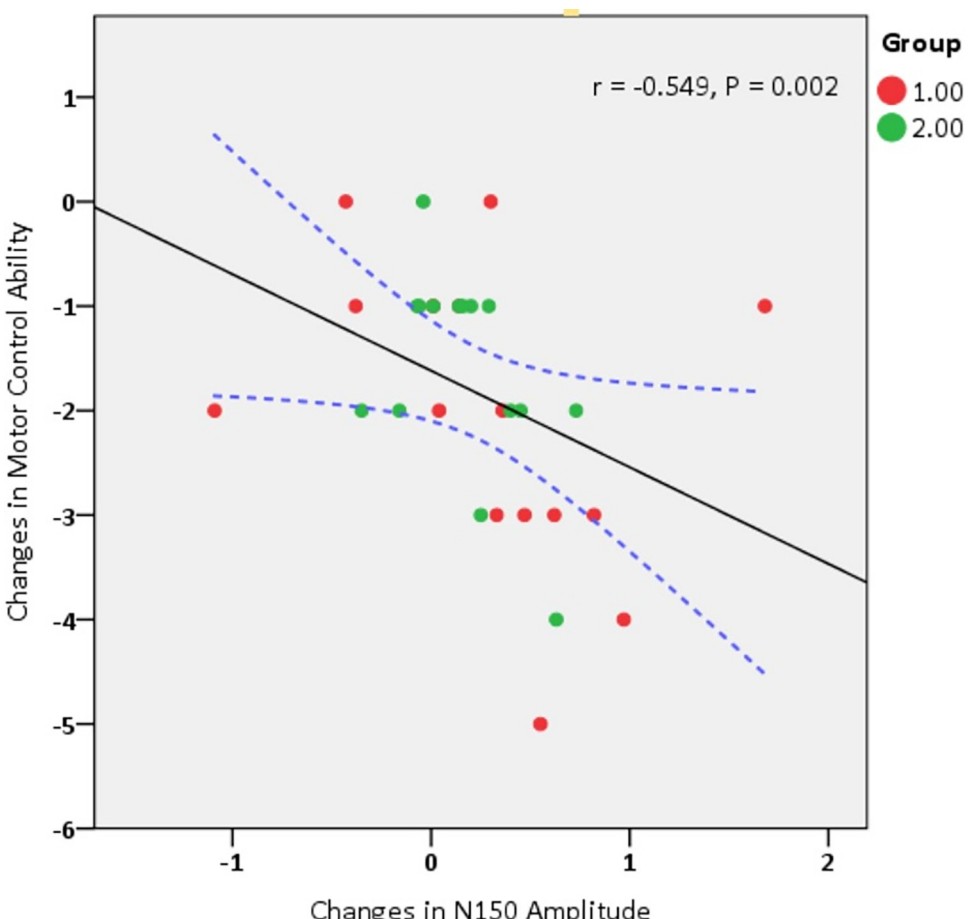

**Fig 4. Correlation between treatment-induced changes in lumbar motor control and N150 amplitude.**

Group 1 represents participants receiving sensory-motor exercises combined with tDCS, while Group 2 includes those receiving sensory-motor exercises combined with sham tDCS.

## 4. Discussion

Both groups showed improvements in pain intensity, disability level, and motor control ability post-treatment compared to pre-treatment. Indeed, the hypothesis of the study was not confirmed in relation to the clinical symptoms. As no current is transmitted during sham electrical stimulation, these observed changes can be attributed to sensory-motor exercises. These exercises involve a broad range of interactions between afferent and efferent signals. The goal of these exercises is to enhance proprioceptive inputs and facilitate appropriate motor responses in dynamic situations. For patients with chronic back pain, sensory-motor exercises aim to improve postural stability, strengthen deep muscles (such as the TrA and MF muscles), and enhance inter-muscular coordination with superficial muscles (such as the oblique abdominal muscles and latissimus) [22]. In this study, sensory-motor exercises involved weight-bearing positions on both legs and one leg, as well as standing on unstable surfaces. Participants' balance was challenged with balls or elastic bands, and muscle contractions in the plantar, abdominal, and deep neck muscles continuously stimulated receptors in the soles, sacroiliac joint, and deep neck muscles. This process helped participants boost proprioceptive inputs, promote coordinated movement patterns, and adjust their posture effectively. Such improvements may manifest as better performance in the Luomajoki test, suggesting significant enhancement in back motor control.

In this regard, Dehbozorgi 's research demonstrated significant enhancement in lumbar motor control (as measured by the Luomajoki test) following sensory-motor exercises using the Huber device, along with a notable reduction in pain compared to the control group [55]. Although another study did not clinically assess the effect of sensory-motor exercises on motor control tests, their efficacy can be inferred from improvement in EMG activity, especially the TrA muscle. It was theorized that sensory-motor exercises aid in postural control strategies and muscle coordination, thereby enhancing motor control in assessments like the Luomajoki test. In line with this, Hwang et al. observed a reduction in activation time of the TrA and external oblique following sensory-motor exercises [56]. In addition, Marchand et al. noted improved activation of the TrA muscle even after just two weeks of training in CLBP patients [57]. They interpreted that sensory-motor exercises improve neuromuscular function and inter-muscular and intramuscular coordination by more efficiently utilizing motor units and increasing the frequency of neurotransmissions [57].

Regarding the reduction of pain and disability in both treatment groups, it can be concluded that sensory-motor exercises play a significant role. As mentioned earlier, the MF and TrA muscles are crucial for dynamic spinal control. The TrA stabilizes the spine by exerting tension on the thoracolumbar fascia and contracting simultaneously with the MF muscle [57]. This study focused on training these muscles through targeted motor control exercises. The findings suggest that such training enhances the dynamic stability of the spine, enabling the body to better handle environmental disturbances, reducing their detrimental effects, and ultimately leading to a decrease in pain and disability. In addition, the improvement in symptoms is likely due to biochemical changes in the blood, such as increased endorphin levels following exercise [58]. Moreover, researchers have noted that exercise enhances corticothalamic excitability and may boost the inhibitory activity of thalamic areas involved in pain regulation, thus effectively reducing pain. Additionally, exercise is associated with increased motivation and pleasure, altering the dopaminergic pathway and internal opioid-producing systems, which modulate pain perception [59]. The findings of this study are consistent with those of

McCaskey [21], Kanabar [60], Wand [5], Wippert [61], and Hwang [56], who all reported significant reductions in pain and disability following sensory-motor exercises.

To the best of our knowledge, no previous studies have investigated the effects of combining sensory-motor exercises with tDCS on pain, disability, and motor control ability in patients with chronic unilateral LSR secondary to L4/L5 and L5/S1 disc herniation. Only two studies have investigated the effects of tDCS in conjunction with training in patients with CLBP [34,35]. The findings of the current study do not align with those of previous research that reported enhanced outcomes from combining tDCS with exercise. For instance, Jafarzadeh's study indicated that combining M1 tDCS with two weeks of postural training effectively alleviated back pain in patients with postural disorders, whereas neither sham tDCS nor postural training alone resulted in significant improvements [34]. In contrast, our study's control group, which received only exercise, still demonstrated improvement.

Several factors may contribute to this discrepancy. First, the duration of the exercises differed significantly; our study lasted four weeks (12 sessions) with each session lasting 40 minutes, compared to Jafarzadeh's two-week intervention (6 sessions) with each session lasting only 20 minutes. Additionally, the types of exercises employed in the two studies varied. Jafarzadeh's research utilized postural training on the Biodex Balance System, where participants stood on a platform, while our study incorporated sensory-motor training. This broader range of movement and proprioceptive challenges in sensory-motor training may provide more comprehensive benefits compared to the more focused postural training in their research. The sensory-motor exercises in our study involved dynamic activities designed to enhance proprioceptive input and muscle coordination. This complexity likely engages both the central and peripheral nervous systems to a greater extent, potentially leading to improved clinical outcomes.

Furthermore, Straudi et al. demonstrated that M1 tDCS, when combined with strengthening exercises, produced significantly better results in pain reduction and psychological well-being compared to sham tDCS [35]. In their study, participants with non-specific CLBP received five sessions of brain stimulation followed by ten sessions of group exercise. Notably, tDCS and exercise were performed during distinct sessions, whereas in our research, both interventions were delivered within the same session. This variation in intervention timing could influence the interaction between brain stimulation and exercise, potentially impacting the overall efficacy of the treatment.

Additionally, our emphasis on deep stabilizing muscles, such as the TrA and MF, through targeted motor control exercises, is likely crucial for enhancing dynamic spinal control. Previous research has highlighted the importance of these muscles in maintaining stability and function, suggesting that our targeted approach could yield substantial benefits. In conclusion, the observed differences in outcomes may be influenced by variations in treatment protocols, including the duration and nature of exercises, the timing of tDCS application, and the specific characteristics of the low back pain being investigated. Further exploration into these factors may provide deeper insights into optimizing therapeutic strategies.

The hypothesis of the study regarding excitability in the sensory and motor cortex was confirmed. Indeed, the results demonstrated that the combination of tDCS with exercises enhances brain excitability (increase in N80 amplitude and decrease in AMT of TrA/IO and MF muscles). Anodal tDCS modifies the resting membrane potential of brain neurons, reducing the threshold for membrane depolarization and increasing neural firing, thereby enhancing cortical excitability for up to 24 hours [35]. Moreover, studies have suggested that anodal tDCS increases local blood flow, while cathodal decreases it [62]. There is also evidence that S1 tDCS increases the amplitude of sensory-evoked potential [63] and enhances somatosensory function in both healthy individuals [64,65] and stroke patients [66]. Therefore, it can be interpreted that adding brain stimulation to the exercises augments cerebral cortex excitability.

Boosting motor cortex excitability was expected to enable modulation of the corticospinal pathway and enhance spinal cord excitability, thereby increasing the transmission of nerve signals to stabilizer muscles and improving back motor control. However, despite a greater increase in brain excitability among participants in the intervention group, improved outcomes for back motor control were not evident. This might be explained by the ceiling effect, indicating that the efficacy of sensory-motor exercises was already substantial enough. Consequently, the addition of tDCS, despite boosting brain excitability, did not confer any additional advantage in clinical symptoms.

Additionally, studies have indicated that one-third of the fibers of the corticospinal tract involved in movement originate from S1 [67], and damage to this area can prevent the acquisition of new motor tasks [68]. Previous research has shown that stimulating S1 with tDCS or TMS improves motor performance in healthy individuals [69], and stroke patients [70]. Conversely, inhibiting S1 with low frequency rTMS in healthy subjects reduces motor skill acquisition [71]. These findings suggest that stimulating S1, through its interaction with M1, can modulate the corticospinal tract, increase the excitability of the spinal network, and facilitate muscle coordination, resulting in the improvement of motor control. In this regard, correlation findings revealed that higher N150 amplitude after treatment corresponds to better performance in the motor control test. The N150 amplitude reflects processing in the S2 [13], which is considered crucial for sensory-motor integration due to its connections with motor areas [12].

It is worth noting that the MEP amplitude of the MF and N150 amplitude significantly increased from pre-test to post-test in both groups, suggesting that the exercises play a crucial role in enhancing cortical excitability. While exercising, the brain constantly processes sensory inputs, which are essential for generating motor outputs. Studies indicate that coherence between sensory inputs and motor performance is instrumental in enhancing motor learning [23]. Consequently, this process can promote motor learning and neuroplastic changes, leading to increased excitability in the motor cortex. Furthermore, growing evidence suggests that physical activity modulates various neurotransmitters, including dopamine, serotonin, norepinephrine, and acetylcholine, along with several neurotrophic factors like brain-derived neurotrophic factor and insulin-like growth factor. These molecules are involved in neurogenesis, synaptogenesis, and angiogenesis, thereby influencing neuroplasticity and potentially altering brain excitability [72]. In this regard, Bae's study has shown that sensory-motor exercises are able to create changes in the brain, thereby enhancing muscle functions and reducing pain [24]. However, this study was limited to seven participants and included 16 treatment sessions. Therefore, further studies are required for a more precise understanding of this issue.

## 5. Limitations

One significant limitation of this study is the absence of control groups, such as those receiving only sensory-motor exercises, solely tDCS, or sham tDCS. The inclusion of these control groups would have enhanced our understanding of the results and their implications. Without them, our ability to accurately evaluate the real clinical effects of the interventions is limited. Specifically, it becomes challenging to assess the individual and combined effects of the interventions. This lack of comparative data obstructs our understanding of how effective each intervention is when analyzed independently. Additionally, the durability of the therapeutic effects was not evaluated, leaving open the possibility that adding tDCS could enhance long-term effects compared to sham. Moreover, a longitudinal study with weekly assessments was not conducted. Implementing such a study could allow for observing the progression of changes and estimating the optimal time to achieve the desired results. This study was

conducted on participants with chronic unilateral LSR secondary to L4/L5 and L5/S1 disc herniation; therefore, the results may not be generalizable to other populations. Moreover, the study tested the effects of tDCS and exercise on numerous outcome measures, along with a wide range of correlation analyses. This substantially increases the risk of Type I errors, thereby heightening the possibility of false-positive findings due to the extensive number of tests performed. Additionally, the sample size was small diminishing the statistical power.

## 6. Conclusion

The findings showed that both sensory-motor exercises combined with either tDCS or sham tDCS effectively reduced pain intensity, decreased disability levels, and improved lumbar motor control in patients with LSR. There were no significant differences in clinical outcomes between the two groups, indicating that adding tDCS did not provide a clear clinical benefit over exercises alone. However, both groups exhibited significant increases in N150 amplitude and MEP amplitude of the MF, indicating enhanced cortical excitability in motor and sensory regions. While the clinical outcomes were similar, the neurophysiological results suggest that sensory-motor exercises play a key role in enhancing cortical excitability. The addition of tDCS further amplified this effect, as shown by a significant reduction in AMT of the MF and TrA/IO muscles and an increase in N80 amplitude.

## 7. Implications for clinical application

The findings support the use of sensory-motor exercises as an effective intervention for reducing pain, improving disability, and enhancing motor control in CLBP patients, while also improving cortical excitability, as evidenced by increased N150 and MEP amplitudes in both groups. Clinicians might consider incorporating sensory-motor exercises as a key component of rehabilitation programs for CLBP. Although the additional benefit of tDCS on clinical outcomes was not evident, its potential to further enhance cortical excitability could be relevant for patients needing more targeted neurophysiological improvements. Future studies could explore whether these neurophysiological changes translate into sustained functional gains or faster recovery.

## Supporting information

**S1 Checklist. CONSORT 2010 checklist of information to include when reporting a randomised trial\*.**
(DOC)

**S1 File. English protocol.**
(DOCX)

**S2 File. Original protocol.**
(DOCX)

## Acknowledgments

We would like to thank the Iranian National Brain Mapping Laboratory (NBML), Tehran, Iran, for their collaboration throughout the study.

## Author Contributions

**Conceptualization:** Soheila Qanbari, Roya Khanmohammadi, Gholamreza Olyaei.

**Data curation:** Soheila Qanbari, Zohreh Hosseini, Hanie Sadat Hejazi.

**Formal analysis:** Soheila Qanbari, Roya Khanmohammadi, Gholamreza Olyaei.

**Funding acquisition:** Roya Khanmohammadi.

**Methodology:** Roya Khanmohammadi.

**Project administration:** Roya Khanmohammadi.

**Supervision:** Gholamreza Olyaei.

**Writing – original draft:** Soheila Qanbari, Zohreh Hosseini, Hanie Sadat Hejazi.

**Writing – review & editing:** Roya Khanmohammadi, Gholamreza Olyaei.

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
