## [Decision Letter · Decision Letter 0]

1 Sep 2024

PONE-D-24-30889Effects of combining sensory-motor exercises with transcranial direct current stimulation on cortical processing and clinical symptoms in patients with chronic low back pain: A sensory and motor-evoked potential studyPLOS ONE

Dear Dr. Khanmohammadi,

Thank you for submitting your manuscript to PLOS ONE. After careful consideration, we feel that it has merit but does not fully meet PLOS ONE’s publication criteria as it currently stands. Therefore, we invite you to submit a revised version of the manuscript that addresses the points raised during the review process.

We look forward to receiving your revised manuscript.

Kind regards,

Abdolvahed Narmashiri

Academic Editor

PLOS ONE

Journal Requirements:

1. When submitting your revision, we need you to address these additional requirements. Please ensure that your manuscript meets PLOS ONE's style requirements, including those for file naming. The PLOS ONE style templates can be found at https://journals.plos.org/plosone/s/file?id=wjVg/PLOSOne_formatting_sample_main_body.pdf and https://journals.plos.org/plosone/s/file?id=ba62/PLOSOne_formatting_sample_title_authors_affiliations.pdf 2. Thank you for stating the following financial disclosure: "This project was funded by the Tehran University of Medical Sciences (Grant No. 1401-2-103-55875)." Please state what role the funders took in the study.  If the funders had no role, please state: "The funders had no role in study design, data collection and analysis, decision to publish, or preparation of the manuscript."If this statement is not correct you must amend it as needed.Please include this amended Role of Funder statement in your cover letter; we will change the online submission form on your behalf. 3. In the online submission form, you indicated that "The data that support the findings of this study are available on request from the corresponding author via Email address". All PLOS journals now require all data underlying the findings described in their manuscript to be freely available to other researchers, either 1. In a public repository, 2. Within the manuscript itself, or 3. Uploaded as supplementary information.This policy applies to all data except where public deposition would breach compliance with the protocol approved by your research ethics board. If your data cannot be made publicly available for ethical or legal reasons (e.g., public availability would compromise patient privacy), please explain your reasons on resubmission and your exemption request will be escalated for approval. 4. Please include captions for your Supporting Information files at the end of your manuscript, and update any in-text citations to match accordingly. Please see our Supporting Information guidelines for more information: http://journals.plos.org/plosone/s/supporting-information.

Additional Editor Comments:

Please address reviewers' comments.

Reviewers' comments:

Reviewer's Responses to Questions

**Comments to the Author**

1. Is the manuscript technically sound, and do the data support the conclusions?

Reviewer #1: Yes

Reviewer #2: Partly

2. Has the statistical analysis been performed appropriately and rigorously? 

Reviewer #1: No

Reviewer #2: Yes

3. Have the authors made all data underlying the findings in their manuscript fully available?

Reviewer #1: Yes

Reviewer #2: No

4. Is the manuscript presented in an intelligible fashion and written in standard English?

Reviewer #1: Yes

Reviewer #2: Yes

5. Review Comments to the Author

Reviewer #1: A two-arm randomized controlled clinical trial was conducted which aimed to explore whether enhancing the excitability of M1 and S1 using tDCS in conjunction with sensory motor exercises provides additional benefit for chronic low back pain patients. The conclusion are unclear.

Major revisions:

Since the group sizes are small (<30) use nonparametric statistical methods for all analyses. Use mixed linear regression models instead of ANOVAs for testing the interactions and/or main effects. A comprehensive reanalysis is needed.

Minor revisions:

1- If the interaction effect is significant, provide an interpretation of the results, but do not test main effects because the tests for main effects are uninteresting in light of significant interactions. If interaction effects are non-significant, drop the interaction effects from the model and test the main effects. Determining which results to present when testing interactions is often a multi-step process.

2- The p-value associated with a correlation is a test of the null hypothesis: correlation equal to zero; however, the absolute magnitude of the coefficient indicates the strength of the linear relationship between two variables. In general, the strength or correlation coefficient is the more important statistic to focus on.

Below is a table for interpreting correlation coefficients:

Coefficient (absolute value) Interpretation

0.90 - 1.0 Very Strong

0.70 - 0.89 Strong

0.40 - 0.69 Moderate

0.10 - 0.39 Weak

less than 0.10 Negligible correlation

3- Cite the statistical software used for the analysis.

4- Table 1: In addition to the frequencies, state the percentages of females and right side pain.

5- Table 1: Indicate the statistical method(s) used to estimate the p-values for sex and side of pain.

6- In the abstract, briefly state the statistical methods from which p-values were obtained.

Reviewer #2: PONE-D-24-30889 : Effects of combining sensory-motor exercises with transcranial direct current stimulation on cortical processing and clinical symptoms in patients with chronic low back pain: A sensory and motor-evoked potential study

General comments:

This randomised controlled trial (RCT) by Qanbari et al. tested the effect of combining tDCS with sensorimotor exercises in participants with lumbar radiculopathy compared to sensorimotor exercises alone on pain, disability, motor control and M1/S1 excitability. They observed that the combination of tDCS and sensorimotor exercises did not influence pain and disability more than sensorimotor exercises. Nonetheless, some neurophysiological variables (such as the motor threshold and SEPs amplitude) can be influenced more by tDCS and exercises than exercises alone. They also observed some correlations between changes in motor control and changes neurophysiological variables. Generally, I appreciate reading this study. Here are some methodological issues that need consideration:

- This study should be referred as an exploratory clinical trial. The sample size is small which limits the interpretation of the results that can be done.

- Please change the title to indicate that the population recruited are patients with lumbar radiculopathy instead of low back pain, and change « a sensory and motor-evoked study” for “an exploratory randomised controlled trial”.

- Authors used a per protocol analyses. This type of analysis increases the likelihood to observed differences between groups. Although it is correct to use such analysis to provide real effect of an intervention, providing the results of an intention-to-treat analyses (e.g. by using linear mixed model) can provide data more in line with the reality. An intention-to-treat analysis will include in the analysis all the participants who have been randomised in a study and all participants will be analysed in their allocated group.

- Authors used Pearson’s correlations without testing the assumptions for using parametric testing. Outliers can largely influence the results of parametric testing and I am concerned that the significant correlation between change in AMT and change in motor control is driven by two participants. I would suggest to provide the information about the normality of the dataset and the presence of outliers or the use Spearman’s correlation. Also, for transparency, I encourage the authors the mention the number of correlations that were tested. Considering that they mentioned that the 3 clinical variables were correlated with the 6 neurophysiological variables, I expect that 18 correlation analyses were computed. I suggest to add a Table with the results (r and p values) for all these analyses. I would be also interesting to identify which the group of each data point in the graph (e.g. using different colours) and adding the 95% confidence interval lines.

Specific comments:

Introduction

p. 3; ln 72-75: I would nuance a bit this sentence. There is no evidence that low back pain is purely neurophysiological either. It is probably a combination of the different factors. In addition, the reference is a narrative review. I suggest citing some original papers suggesting alteration in CNS.

p. 3-4; ln 87-94: About this section, although Corti et al. demonstrated changes in M1 excitability, TMS targeted M1 area of finger muscles which differs of the rationale presented by the authors(Corti et al., 2022). Thus, I would remove this reference. Also, Tsao et al. (2011) should be cited because they were the first to smudging of M1 maps of the erector spinae in LBP (Tsao et al., 2011). Finally, a recent paper showed that sensorimotor integration was altered in CLBP i.e. that the sensory afferents from the low back area increased corticospinal excitability compared to pain-free controls (Masse-Alarie et al., 2024). This study may help to relate the S1 and M1 paragraphs of the introduction.

p.4; ln 95-98: I would be careful here. The evidence except the one from Jenkins et al. are all case-control studies. I would rather mention that it is possible that some alterations in sensorimotor control may contribute to LBP.

p. 4; ln 104: change « progression » for « maintenance ».

p. 4; ln 111-114 : Other research also demonstrated that motor control exercise can modify the plasticity of the M1, I invite the authors to refer to them (Massé-Alarie et al., 2016, Tsao et al., 2010).

p.4; ln 115: Again, I would change “CLBP is associated” by “some patients with LBP present with a different organization of S1 and M1 areas […]”

p. 5; ln 125: Please refer to systematic review done in LBP (Alwardat et al., 2020, Patricio et al., 2021) that is more related to the topic of the paper.

p. 5; ln 137: change “neuroimaging” for “neurostimulation” if it refers to TMS mapping.

p. 5; ln 139: remove “improved”

Methods

p. 6; 162-164: Why did the authors choose to recruit participants with lumbar radiculopathy? Most results in the introduction comes from non-specific LBP. It is less clear if patients with radiculopathy presents changes in M1 and S1. I would be important to add in the introduction some lines on the absence of data in radiculopathy. Also, it would be important to refer to an rTMS and tDCS study who did test its effectiveness in participants with radiculopathy (Attal et al., 2016)

p. 6; ln 174: change “L5/L4 » for “L4/L5”.

p. 6; ln 186-189: These are not exclusion criteria because participants were first included then excluded. These criteria should be rather mentioned in the statistical analysis to describe the per protocol criteria (i.e. the criteria that were used to include a participant in the analysis).

p. 7; ln 192-193: Please describe the mean difference and the standard deviation used to calcite the effect size.

p. 7; ln 207-209: Please indicate the timeframe on which pain was measured (current pain, average pain in the last 24 hours, average pain in the last week, worst pain, etc.). If authors assess current pain, please describe the position of the participants when pain was questioned since it may influence the pain intensity.

p. 9; ln 267-269: For trunk muscles, it is very difficult to obtain MEPs of large amplitude. Most published papers report MEPs amplitude below 100 μV that may make impossible to use this as a criterion for motor threshold (e.g. (Desmons et al., 2024, Desmons et al., 2021)). Did it happen to the authors? Were they able to find MEps over 100 μV in most patients?

p. 10; ln 298-301: Please confirm and add to this section that the intervention lasted for 4 weeks.

p. 11; exercises: What is the rationale to train the foot, lumbar and cervical spine? Why did authors emphasize on these body areas while they were testing only M1/S1 excitability of the low back muscles?

p. 12; ln 364-366: Were assumptions for the use of parametric tests done? With a small sample size, outliers can influence a lot the results of Pearson’s correlations.

Results

Table 2: The way the data are presented makes difficult to compare the effects of each intervention. I would suggest presenting the pre-post data for each group side-by-side followed by the post-to-pre mean difference (within-group) with standard deviation or 95% confidence interval. Also, it would be interesting to indicate where there is a significant interaction.

Experimental Control

Pre Post Mean difference Pre Post Mean difference

p. 13; Ln 390-392: Please indicate if there is any difference between groups for pre and post timepoints each time an interaction occurs to determine if the interaction was explained by differences at baseline or at the follow-up timepoints.

p. 14; linear relationships: Please indicate how many correlations were tested. Please indicate that no other correlation was significant. Did any corrections were applied for multiple testing? Is there any significant correlation between change in motor control and change in pain or disability?

Discussion

p. 16; ln 493 change “addressed” for “tested”.

p. 18; Limitations : Please add that the effect of tDCS and exercises were tested on many outcomes. Also, a lot of correlations were tested. This increases the number of type I error. In addition, the sample size was very small.  

Alwardat M, Pisani A, Etoom M, Carpenedo R, Chinè E, Dauri M, et al. Is transcranial direct current stimulation (tDCS) effective for chronic low back pain? A systematic review and meta-analysis. Journal of Neural Transmission 2020:1-14.

Attal N, Ayache SS, Ciampi De Andrade D, Mhalla A, Baudic S, Jazat F, et al. Repetitive transcranial magnetic stimulation and transcranial direct-current stimulation in neuropathic pain due to radiculopathy: a randomized sham-controlled comparative study. Pain 2016;157(6):1224-31.

Corti EJ, Marinovic W, Nguyen AT, Gasson N, Loftus AM. Motor cortex excitability in chronic low back pain. Exp Brain Res 2022;240(12):3249-57.

Desmons M, Cherif A, Rohel A, de Oliveira FCL, Mercier C, Masse-Alarie H. Corticomotor Control of Lumbar Erector Spinae in Postural and Voluntary Tasks: The Influence of Transcranial Magnetic Stimulation Current Direction. eNeuro 2024;11(2).

Desmons M, Rohel A, Desgagnes A, Mercier C, Masse-Alarie H. Influence of different transcranial magnetic stimulation current directions on the corticomotor control of lumbar erector spinae muscles during a static task. J Neurophysiol 2021;126(4):1276-88.

Massé-Alarie H, Beaulieu L, Preuss R, Schneider C. Influence of paravertebral muscles training on brain plasticity and postural control in chronic low back pain. Scandinavian Journal of Pain 2016;12:74-83.

Masse-Alarie H, Shraim M, Hodges PW. Sensorimotor Integration in Chronic Low Back Pain. Neuroscience 2024;552:29-38.

Patricio P, Roy JS, Rohel A, Gariepy C, Emond C, Hamel E, et al. The Effect of Noninvasive Brain Stimulation to Reduce Nonspecific Low Back Pain: A Systematic Review and Meta-analysis. Clin J Pain 2021;37(6):475-85.

Tsao H, Danneels LA, Hodges PW. ISSLS prize winner: Smudging the motor brain in young adults with recurrent low back pain. Spine (Phila Pa 1976) 2011;36(21):1721-7.

Tsao H, Galea MP, Hodges PW. Driving plasticity in the motor cortex in recurrent low back pain. Eur J Pain 2010;14(8):832-9.

6. PLOS authors have the option to publish the peer review history of their article (what does this mean?). If published, this will include your full peer review and any attached files.

Reviewer #1: No

Reviewer #2: No

---

## [Author Response · Author response to Decision Letter 0]

2 Oct 2024

Dear Reviewer 1

We sincerely appreciate your thorough review of the manuscript and the valuable comments and suggestions provided. We believe that the manuscript has significantly improved as a result of incorporating your feedback, which has been highlighted in red throughout the text.

Question 1: A two-arm randomized controlled clinical trial was conducted which aimed to explore whether enhancing the excitability of M1 and S1 using tDCS in conjunction with sensory motor exercises provides additional benefit for chronic low back pain patients. The conclusion is unclear.

Answer 1: We modified the conclusion to “The findings showed that both sensory-motor exercises combined with either tDCS or sham tDCS effectively reduced pain intensity, decreased disability levels, and improved lumbar motor control in patients with lumbosacral radiculopathy. There were no significant differences in clinical outcomes between the two groups, indicating that adding tDCS did not provide a clear clinical benefit over exercises alone. However, both groups exhibited significant increases in N150 amplitude and MEP amplitude of the MF, indicating enhanced cortical excitability in motor and sensory regions. While the clinical outcomes were similar, the neurophysiological results suggest that sensory-motor exercises play a key role in enhancing cortical excitability. The addition of tDCS further amplified this effect, as shown by a significant reduction in AMT of the MF and TrA/IO muscles and an increase in N80 amplitude.”

Question 2: Since the group sizes are small (<30) use nonparametric statistical methods for all analyses. Use mixed linear regression models instead of ANOVAs for testing the interactions and/or main effects. A comprehensive reanalysis is needed.

Answer 2: The reason for not using non-parametric tests was that, despite the small sample size, the data exhibited a normal distribution. However, based on your feedback and that of another reviewer, we used linear mixed effect model (LMM) instead of ANOVA, and the tables were updated accordingly. Although the tables were modified, the final results did not significantly change, so the discussion remains the same.

Moreover, the correlation statistical analysis was adjusted because the treatment- induced changes in outcomes (post-test minus pre-test) did not follow a normal distribution; therefore, we used Spearman's method for this analysis. 

Question 3: If the interaction effect is significant, provide an interpretation of the results, but do not test main effects because the tests for main effects are uninteresting in light of significant interactions. If interaction effects are non-significant, drop the interaction effects from the model and test the main effects. Determining which results to present when testing interactions is often a multi-step process.

Answer 3: Thank you for your comment. We have taken this into account in the interpretation of all the results, as detailed in the results section.

Question 4: The p-value associated with a correlation is a test of the null hypothesis: correlation equal to zero; however, the absolute magnitude of the coefficient indicates the strength of the linear relationship between two variables. In general, the strength or correlation coefficient is the more important statistic to focus on.

Below is a table for interpreting correlation coefficients:

Coefficient (absolute value) Interpretation

0.90 - 1.0 Very Strong

0.70 - 0.89 Strong

0.40 - 0.69 Moderate

0.10 - 0.39 Weak

less than 0.10 Negligible correlation

Answer 4: We added the following sentences to the text.

“The correlation strength was assessed using the correlation coefficient r, which was categorized as follows: a value between 0.90 and 1.0 indicates a very strong correlation; 0.70 to 0.89 signifies a strong correlation; 0.40 to 0.69 represents a moderate correlation; 0.10 to 0.39 suggests a weak correlation; and values less than 0.10 reflect a negligible correlation.”

In addition, we included a table displaying the correlations between the three clinical variables and six neurophysiological variables, incorporating statistical corrections regardless of the significance of the results.

Question 5: Cite the statistical software used for the analysis.

Answer 5: We added the following sentence to the text.

“For statistical analysis, IBM SPSS Statistics version 23.0 was utilized.”

Question 6: Table 1: In addition to the frequencies, state the percentages of females and right side pain.

Answer 6: We included the percentages in the Table 1.

Question 7: Table 1: Indicate the statistical method(s) used to estimate the p-values for sex and side of pain.

Answer 7: The following sentence was added to the text.

“The Chi-square test was applied to compare the frequency of gender and side of pain between the groups.”

Question 8: In the abstract, briefly state the statistical methods from which p-values were obtained.

Answer 8: We added the following sentences to the abstract 

“To assess the main effects of group, time, and their interactions, a linear mixed effects model (LMM) was utilized. Furthermore, to evaluate the relationship between treatment-induced changes (post-test minus pre-test) in neurophysiological and clinical outcomes, Spearman's correlation coefficient was used.”

Dear Reviewer 2

We sincerely appreciate your thorough review of the manuscript and the valuable comments and suggestions provided. We believe that the manuscript has significantly improved as a result of incorporating your feedback, which has been highlighted in blue throughout the text.

Question 1: This study should be referred as an exploratory clinical trial. The sample size is small which limits the interpretation of the results that can be done. Please change the title to indicate that the population recruited are patients with lumbar radiculopathy instead of low back pain, and change « a sensory and motor-evoked study” for “an exploratory randomised controlled trial”.

Answer 1: We changed the title to “Effects of combining sensory-motor exercises with transcranial direct current stimulation on cortical processing and clinical symptoms in patients with lumbar radiculopathy: an exploratory randomised controlled trial”

Question 2: Authors used a per protocol analyses. This type of analysis increases the likelihood to observed differences between groups. Although it is correct to use such analysis to provide real effect of an intervention, providing the results of an intention-to-treat analyses (e.g. by using linear mixed model) can provide data more in line with the reality. An intention-to-treat analysis will include in the analysis all the participants who have been randomised in a study and all participants will be analysed in their allocated group.

Answer 2: We revised our analysis and utilized a linear mixed model instead of two-way mixed repeated measures ANOVA. Consequently, we added the following sentences to the statistical analysis section. Although the tables were modified, the final results did not significantly change, so the discussion remains the same.

“To examine the main effects of group (sensory-motor exercises + tDCS vs. sensory-motor exercises + sham tDCS), time (pre- and post-treatment), and their interactions, a linear mixed effects model (LMM) was performed. When significant interactions were identified, the groups were analyzed separately for more detailed insights. Time, group, and their interaction effects were considered fixed effects, while participants were treated as random effects. Mixed models are generally favored over traditional general linear models for handling missing data, as they can utilize all data points, even when some observations are absent. This model evaluates changes over time using maximum likelihood estimation, thus applying an intention-to-treat approach that yields data more reflective of real-world conditions.”

Question 3: Authors used Pearson’s correlations without testing the assumptions for using parametric testing. Outliers can largely influence the results of parametric testing and I am concerned that the significant correlation between change in AMT and change in motor control is driven by two participants. I would suggest to provide the information about the normality of the dataset and the presence of outliers or the use Spearman’s correlation. Also, for transparency, I encourage the authors the mention the number of correlations that were tested. Considering that they mentioned that the 3 clinical variables were correlated with the 6 neurophysiological variables, I expect that 18 correlation analyses were computed. I suggest to add a Table with the results (r and p values) for all these analyses. I would be also interesting to identify which the group of each data point in the graph (e.g. using different colours) and adding the 95% confidence interval lines.

Answer 3: We revised this section accordingly and included a table that outlines all 18 correlations between clinical and neurophysiological parameters. Moreover, we included 95% confidence interval lines in Figure 4 and distinguished each group in the graph by using different colors.

“Additionally, to examine the relationship between treatment-induced changes (post-test minus pre-test) in neurophysiological and clinical outcomes, Spearman's correlation coefficient was employed, as the change values did not follow a normal distribution. These correlations were computed for all participants regardless of their group assignment. Given that the analysis involved 3 clinical parameters (pain intensity, disability level, and lumbar motor control ability) and 6 neurophysiological parameters (N80 and N150 amplitudes, MEP amplitude, and AMT for multifidus and transversus abdominis/internal oblique muscles), a total of 18 correlations (3×6) were tested. To account for the risk of Type I errors from multiple comparisons, the Bonferroni correction was applied, resulting in a corrected p-value threshold of 0.003.”

Your concern about the relationship between change in AMT and change in motor control was valid, as the revised analysis resulted in a weaker relationship that was no longer statistically significant. We appreciate your careful attention to this matter.

Introduction

Question 4: p. 3; ln 72-75: I would nuance a bit this sentence. There is no evidence that low back pain is purely neurophysiological either. It is probably a combination of the different factors. In addition, the reference is a narrative review. I suggest citing some original papers suggesting alteration in CNS.

Answer 4: We revised this sentence accordingly and included several original studies to support the claim. 

“Indeed, the underlying mechanisms of musculoskeletal disorders are not limited to structural and biomechanical factors; neurophysiological processes also play a significant role in their development and progression [6-9].”

Question 5: p. 3-4; ln 87-94: About this section, although Corti et al. demonstrated changes in M1 excitability, TMS targeted M1 area of finger muscles which differs of the rationale presented by the authors(Corti et al., 2022). Thus, I would remove this reference. Also, Tsao et al. (2011) should be cited because they were the first to smudging of M1 maps of the erector spinae in LBP (Tsao et al., 2011). Finally, a recent paper showed that sensorimotor integration was altered in CLBP i.e. that the sensory afferents from the low back area increased corticospinal excitability compared to pain-free controls (Masse-Alarie et al., 2024). This study may help to relate the S1 and M1 paragraphs of the introduction.

Answer 5: We have removed the Corti et al. study from the references and included the Tsao et al. study. Additionally, we incorporated sentences detailing the findings of the Masse-Alarie et al. study as follow:

“Moreover, a recent study by Masse-Alarie et al. found that sensorimotor integration was disrupted in individuals with CLBP. Specifically, sensory input from the lower back region enhanced corticospinal excitability compared to those without pain. As a result, they recommended interventions aimed at improving both sensory processing and spinal motor control [17].”

Question 6: p.4; ln 95-98: I would be careful here. The evidence except the one from Jenkins et al. are all case-control studies. I would rather mention that it is possible that some alterations in sensorimotor control may contribute to LBP.

Answer 6: Based on your feedback, we revised these sentences to “These findings indicate that changes in sensorimotor integration and decreased excitability in S1 and M1, which disrupt sensorimotor control, may contribute to chronic and recurrent low back pain.”

Question 7: p. 4; ln 104: change « progression » for « maintenance ».

Answer 7: We revised it. 

Question 8: p. 4; ln 111-114 : Other research also demonstrated that motor control exercise can modify the plasticity of the M1, I invite the authors to refer to them (Massé-Alarie et al., 2016, Tsao et al., 2010).

Answer 8: We included the two mentioned studies as references.

Question 9: p.4; ln 115: Again, I would change “CLBP is associated” by “some patients with LBP present with a different organization of S1 and M1 areas […]”

Answer 9: We applied your comment in the text.

Question 10: p. 5; ln 125: Please refer to systematic review done in LBP (Alwardat et al., 2020, Patricio et al., 2021) that is more related to the topic of the paper.

Answer 10: We included these references to the text.

Question 11: p. 5; ln 137: change “neuroimaging” for “neurostimulation” if it refers to TMS mapping.

Answer 11: We modified it.

Question 12: p. 5; ln 139: remove “improved”

Answer 12: We removed “improved”.

Method

Question 13: p. 6; 162-164: Why did the authors choose to recruit participants with lumbar radiculopathy? Most results in the introduction comes from non-specific LBP. It is less clear if patients with radiculopathy presents changes in M1 and S1. I would be important to add in the introduction some lines on the absence of data in radiculopathy. Also, it would be important to refer to an rTMS and tDCS study who did test its effectiveness in participants with radiculopathy (Attal et al., 2016)

Answer 13: Based on your feedback, we included the following paragraphs to the introduction

“Chronic low back pain (CLBP) is one of the most prevalent health issues worldwide, affecting up to 80% of individuals, with 5% to 10% suffering lumbosacral radiculopathy [1]. This condition is characterized by radicular pain resulting from the compression or irritation of nerve roots in the lumbosacral region of the spine [2]. CLBP is a multifaceted sensory and emotional condition influenced by mechanical, psychological, and psychosocial factors [3]. Consequently, many patients experience persistent radicular symptoms, and some may require surgical intervention. However, not all patients achieve satisfactory pain relief, even with appropriate treatments [4]. Thus, the underlying cause of persistent pain remains unclear. Evidence indicates that treatments focused solely on correcting structural abnormalities in the musculoskeletal system are often unsuccessful, suggesting the involvement of additional pathophysiological and biopsychosocial mechanisms [5].”

“It is important to note that most of these findings are derived from non-specific CLBP, while research on changes in the M1 and S1 in patients with lumbosacral radiculopathy is limited. Nevertheless, studies have indicated that brain networks are also affected in these patients [4, 19].”

Additionally, a study by Attal et al. on patients with lumbosacral radiculopathy found that repetitive transcranial magnetic stimulation (rTMS) over M1 was more effective than tDCS and sham treatments, potentially modulating both the sensory and affective aspects of pain [34].

Question 14: p. 6; ln 174: change “L5/L4 » for “L4/L5”.

Answer 14: We corrected it. 

Question 15: p. 6; ln 186-189: These are not exclusion criteria because participants were first included then excluded. These criteria should be rather mentioned in the statistical analysis to describe the per protocol criteria (i.e. the criteria that were used to include a participant in the analysis).

Answer 15: We removed this section because, as you 

---

## [Decision Letter · Decision Letter 1]

29 Oct 2024

PONE-D-24-30889R1Effects of combining sensory-motor exercises with transcranial direct current stimulation on cortical processing and clinical symptoms in patients with lumbosacral radiculopathy: an exploratory randomized controlled trialPLOS ONE

Dear Dr. Khanmohammadi,

Thank you for submitting your manuscript to PLOS ONE. After careful consideration, we feel that it has merit but does not fully meet PLOS ONE’s publication criteria as it currently stands. Therefore, we invite you to submit a revised version of the manuscript that addresses the points raised during the review process.

We look forward to receiving your revised manuscript.

Kind regards,

Abdolvahed Narmashiri

Academic Editor

PLOS ONE

Journal Requirements:

**Additional Editor Comments:**

Thank you to the authors for addressing all the reviewer's comments. Although the manuscript has improved after revision, there are still a few minor points to address.

-The abstract is lengthy and could be more focused. Please rewrite to include: a brief statement of the research gap, a summary of the methodology, primary findings, and a concise conclusion.

-Introduction sometimes provides excessive detail, especially regarding studies on related neuroanatomical structures and findings that might detract from the central aims. A more concise overview could be achieved by summarizing these points and focusing on studies most relevant to the proposed intervention.

-The paragraph transitions are generally clear, but the introduction could benefit from improved cohesion between sections. For instance, while the background on CLBP and the CNS’s role is essential, the connection to tDCS and sensory-motor interventions could be introduced earlier to set the stage for later sections.

-Additionally, the shift from discussing broad CLBP pathophysiology to specifics about M1 and S1 activity lacks a clear transition. Structuring the introduction to start with general context, then gradually narrow down to neurophysiological mechanisms and intervention-specific findings, would improve readability.

-Although the introduction identifies limitations in previous studies, the justification for targeting both M1 and S1 with tDCS should be articulated more explicitly. While it mentions that most studies focus on M1 alone, it could be beneficial to briefly elaborate on why stimulating both regions might be superior in treating CLBP.

-The research gap around sensory-motor integration is mentioned but could be emphasized further. Highlighting specific gaps, such as the lack of studies on combined sensory and motor cortical stimulation in CLBP patients, would strengthen the justification for this study.

-The writing is mostly clear, but there are minor grammatical improvements needed for a polished presentation.

-Avoid redundancy, such as using “in patients with lumbosacral radiculopathy” multiple times in close proximity. Rephrase or consolidate similar statements.

-The methods and results are well-organized.

-The discussion is well-organized but would benefit from clearer segmentation between clinical implications, neurophysiological insights, and study limitations. Grouping related ideas together could improve readability.

-Some sentences are dense, making it challenging to follow the main argument. Breaking complex information into shorter, direct statements would enhance comprehension.

-The comparison with prior studies, especially regarding differing effects of tDCS and the nature of exercises, is valuable but could be extended by exploring why the results diverged from these studies.

Reviewers' comments:

Reviewer's Responses to Questions

**Comments to the Author**

1. If the authors have adequately addressed your comments raised in a previous round of review and you feel that this manuscript is now acceptable for publication, you may indicate that here to bypass the “Comments to the Author” section, enter your conflict of interest statement in the “Confidential to Editor” section, and submit your "Accept" recommendation.

Reviewer #1: (No Response)

Reviewer #2: All comments have been addressed

2. Is the manuscript technically sound, and do the data support the conclusions?

Reviewer #1: Yes

Reviewer #2: Yes

3. Has the statistical analysis been performed appropriately and rigorously? 

Reviewer #1: Yes

Reviewer #2: Yes

4. Have the authors made all data underlying the findings in their manuscript fully available?

Reviewer #1: Yes

Reviewer #2: No

5. Is the manuscript presented in an intelligible fashion and written in standard English?

Reviewer #1: Yes

Reviewer #2: Yes

6. Review Comments to the Author

Reviewer #1: Minor revisions:

1- Table 1: Check the data for normal distributions. If variables do not follow normal distributions, summarize using median, first and third quartiles and compare using Wilcoxon rank sum test.

2- Lines 250 & 314: The standard statistical term for “average” is “mean”.

3- Line 401: The chi-square test compares proportions rather than frequencies.

Reviewer #2: PONE-D-24-30889_R1: Effects of combining sensory-motor exercises with transcranial direct current stimulation on cortical processing and clinical symptoms in patients with lumbosacral radiculopathy: an exploratory randomized controlled trial

I would like to thank the Authors for considering my comments of the previous review. I have only minor comments.

Introduction

Ln 170: Change “andal” for “anodal”

Methods

Ln 415; ln 417-423: Please add references for the benchmarks used.

Results

For post-hoc comparisons, was there any differences at post time for outcomes presenting with interaction effects)? This information was provided for all AMT of TrA but not for AMT of MF and N80.

Discussion

Ln 524: Reference 49 should be 54

Ln 529-530: I would change the term “strengthening” for “training”. I don’t think the current protocol allows to strengthen these muscles, at least, this is not the objective of motor control exercise.

Ln 530-540: Authors need to be careful, there is no group with a placebo or a “no intervention” limiting the interpretability of the effect of exercise. It needs to be considered in the discussion. For example, it could be added in Ln 612-614 that the absence of these groups limits to determine the real clinical effects of the two groups tested.

7. PLOS authors have the option to publish the peer review history of their article (what does this mean?). If published, this will include your full peer review and any attached files.

Reviewer #1: No

Reviewer #2: No

---

## [Author Response · Author response to Decision Letter 1]

2 Nov 2024

Dear Editor

We sincerely appreciate your thorough review of the manuscript and the valuable comments and suggestions provided. We believe that the manuscript has significantly improved as a result of incorporating your feedback, which has been highlighted in green throughout the text.

Question 1: The abstract is lengthy and could be more focused. Please rewrite to include: a brief statement of the research gap, a summary of the methodology, primary findings, and a concise conclusion.

Answer 1: We shortened the abstract from 442 to 369 words. Given the journal's 500-word limit, we chose to present the results with the desired level of detail.

Question 2: Introduction sometimes provides excessive detail, especially regarding studies on related neuroanatomical structures and findings that might detract from the central aims. A more concise overview could be achieved by summarizing these points and focusing on studies most relevant to the proposed intervention.

Answer 2: We attempted to eliminate a few sentences; however, removing more is not feasible, as these studies offer substantial evidence regarding changes in the sensory and motor areas of the brain in individuals with back pain. Furthermore, several of them were recommended by the reviewers to strengthen the rationale of the study.

Question 3: The paragraph transitions are generally clear, but the introduction could benefit from improved cohesion between sections. For instance, while the background on CLBP and the CNS’s role is essential, the connection to tDCS and sensory-motor interventions could be introduced earlier to set the stage for later sections.

Answer 3: We added the following sentences to the 5th paragraph “…….They recommended interventions targeting both sensory processing and spinal motor control [17]. These interventions could include exercises that promote top-down influences on sensorimotor integration or brain stimulation techniques that directly target these pathways.”

Question 4: Additionally, the shift from discussing broad CLBP pathophysiology to specifics about M1 and S1 activity lacks a clear transition. Structuring the introduction to start with general context, then gradually narrow down to neurophysiological mechanisms and intervention-specific findings, would improve readability.

Answer 4: We added the following sentences to improve transition between paragraphs “Understanding the neurophysiological mechanisms associated with CLBP is essential for addressing the sensory-motor integration deficits linked to this condition. A critical component of this understanding involves the sensory cortex (S1) and the primary motor cortex (M1), which both show significant changes in individuals with CLBP…………” And “In light of these findings, the M1 also emerges as a critical area of focus. Studies by……….”

Question 5: Although the introduction identifies limitations in previous studies, the justification for targeting both M1 and S1 with tDCS should be articulated more explicitly. While it mentions that most studies focus on M1 alone, it could be beneficial to briefly elaborate on why stimulating both regions might be superior in treating CLBP. The research gap around sensory-motor integration is mentioned but could be emphasized further. Highlighting specific gaps, such as the lack of studies on combined sensory and motor cortical stimulation in CLBP patients, would strengthen the justification for this study.

Answer 5: We modified this section to “While existing studies provide valuable insights, they primarily focus on stimulating the M1 and overlook the fact that individuals with CLBP also exhibit reduced excitability in the S1. Given the alterations in both regions and S1's crucial role in sensory integration and pain processing, targeting both M1 and S1 may offer a more comprehensive and effective strategy for alleviating CLBP symptoms.

Despite the recognition of sensory-motor integration deficits as a significant factor in CLBP, research on interventions targeting both sensory and motor cortices remains limited. Combining tDCS of these areas could enhance the efficacy of sensory-motor exercises by improving excitability in both domains, potentially leading to more substantial relief in CLBP management.

Therefore, this study aims to address this gap by investigating whether enhancing excitability in M1 and S1 through anodal tDCS, in conjunction with sensory-motor exercises, can improve sensory-motor brain processing. This approach may subsequently alleviate pain, reduce disability, and enhance lumbar motor control in patients with CLBP. To our knowledge, no previous studies have explored the combined effects of M1 and S1 stimulation with sensory-motor exercises on clinical symptoms in individuals with LSR.”

Question 6: The writing is mostly clear, but there are minor grammatical improvements needed for a polished presentation.

Answer 6: The text has been revised by a native English speaker.

Question 7: Avoid redundancy, such as using “in patients with lumbosacral radiculopathy” multiple times in close proximity. Rephrase or consolidate similar statements.

Answer 7: The participants in this study are individuals with lumbosacral radiculopathy. To avoid excessive repetition of this term throughout the text, we have assigned the abbreviation 'LSR' for convenience.

Question 8: The discussion is well-organized but would benefit from clearer segmentation between clinical implications, neurophysiological insights, and study limitations. Grouping related ideas together could improve readability.

Answer 8: As you can see, in this study, we have addressed the clinical implications and study limitations in two distinct sections titled "Implications for Clinical Application" and "Limitations." Furthermore, the discussion includes an exploration of potential neurophysiological mechanisms and comparisons of our results with findings from other studies.

Question 9: Some sentences are dense, making it challenging to follow the main argument. Breaking complex information into shorter, direct statements would enhance comprehension.

Answer 9: We tried to simplify complex information by breaking it down into shorter segments.

Question 10: The comparison with prior studies, especially regarding differing effects of tDCS and the nature of exercises, is valuable but could be extended by exploring why the results diverged from these studies.

Answer 10: We modified this section to “To the best of our knowledge, no previous studies have investigated the effects of combining sensory-motor exercises with tDCS on pain, disability, and motor control ability in patients with chronic unilateral LSR secondary to L4/L5 and L5/S1 disc herniation. Only two studies have investigated the effects of tDCS in conjunction with training in patients with CLBP [34, 35]. The findings of the current study do not align with those of previous research that reported enhanced outcomes from combining tDCS with exercise. For instance, Jafarzadeh's study indicated that combining M1 tDCS with two weeks of postural training effectively alleviated back pain in patients with postural disorders, whereas neither sham tDCS nor postural training alone resulted in significant improvements [34]. In contrast, our study's control group, which received only exercise, still demonstrated improvement. 

Several factors may contribute to this discrepancy. First, the duration of the exercises differed significantly; our study lasted four weeks (12 sessions) with each session lasting 40 minutes, compared to Jafarzadeh's two-week intervention (6 sessions) with each session lasting only 20 minutes. Additionally, the types of exercises employed in the two studies varied. Jafarzadeh's research utilized postural training on the Biodex Balance System, where participants stood on a platform, while our study incorporated sensory-motor training. This broader range of movement and proprioceptive challenges in sensory-motor training may provide more comprehensive benefits compared to the more focused postural training in their research. The sensory-motor exercises in our study involved dynamic activities designed to enhance proprioceptive input and muscle coordination. This complexity likely engages both the central and peripheral nervous systems to a greater extent, potentially leading to improved clinical outcomes. 

Furthermore, Straudi et al. demonstrated that M1 tDCS, when combined with strengthening exercises, produced significantly better results in pain reduction and psychological well-being compared to sham tDCS [35]. In their study, participants with non-specific CLBP received five sessions of brain stimulation followed by ten sessions of group exercise. Notably, tDCS and exercise were performed during distinct sessions, whereas in our research, both interventions were delivered within the same session. This variation in intervention timing could influence the interaction between brain stimulation and exercise, potentially impacting the overall efficacy of the treatment.

Additionally, our emphasis on deep stabilizing muscles, such as the TrA and MF, through targeted motor control exercises, is likely crucial for enhancing dynamic spinal control. Previous research has highlighted the importance of these muscles in maintaining stability and function, suggesting that our targeted approach could yield substantial benefits. In conclusion, the observed differences in outcomes may be influenced by variations in treatment protocols, including the duration and nature of exercises, the timing of tDCS application, and the specific characteristics of the low back pain being investigated. Further exploration into these factors may provide deeper insights into optimizing therapeutic strategies.”

Dear Reviewer 1

We sincerely appreciate your thorough review of the manuscript and the valuable comments and suggestions provided. We believe that the manuscript has significantly improved as a result of incorporating your feedback, which has been highlighted in red throughout the text.

Question 1: 1- Table 1: Check the data for normal distributions. If variables do not follow normal distributions, summarize using median, first and third quartiles and compare using Wilcoxon rank sum test 

Answer 1: We reviewed and revised them in Table 1. Furthermore, the related text was modified accordingly.

 “The Shapiro-Wilk test was used to evaluate the normality of the data distribution. Most data followed a normal distribution, except for certain demographic characteristics like height, weight, and pain duration. The Chi-square test was used to compare the proportions of gender and pain side between the groups. For parameters with a normal distribution, an independent t-test was conducted, while the Wilcoxon rank-sum test was applied to non-normally distributed parameters. The results indicated no significant baseline differences between the two groups, confirming their comparability before the intervention.”

Question 2: Lines 250 & 314: The standard statistical term for “average” is “mean”.

Answer 2: We revised it. 

Question 3: Line 401: The chi-square test compares proportions rather than frequencies.

Answer 3: We revised it. 

Dear Reviewer 2

We sincerely appreciate your thorough review of the manuscript and the valuable comments and suggestions provided. We believe that the manuscript has significantly improved as a result of incorporating your feedback, which has been highlighted in blue throughout the text.

Question 1: Introduction

Ln 170: Change “andal” for “anodal”

Answer 1: We revised it.

Question 2: Methods

Ln 415; ln 417-423: Please add references for the benchmarks used.

Answer 2: We added the related references to the text.

Question 3: Results

For post-hoc comparisons, was there any differences at post time for outcomes presenting with interaction effects? This information was provided for all AMT of TrA but not for AMT of MF and N80.

Answer 3: The group effect was significant only for the AMT of TrA/IO, whereas it was not significant for the AMT of MF and N80, indicating no differences between groups at the post-test for these two parameters. We have updated the text to reflect that the group effect was not significant for these measures.

Question 4: Ln 524: Reference 49 should be 54

Answer 4: We revised it. 

Question 5: Ln 529-530: I would change the term “strengthening” for “training”. I don’t think the current protocol allows to strengthen these muscles, at least, this is not the objective of motor control exercise.

Answer 5: We changed “strengthening” to “training”.

Question 6: Ln 530-540: Authors need to be careful, there is no group with a placebo or a “no intervention” limiting the interpretability of the effect of exercise. It needs to be considered in the discussion. For example, it could be added in Ln 612-614 that the absence of these groups limits to determine the real clinical effects of the two groups tested.

Answer 6: We have incorporated the following text in response to your feedback.

“One significant limitation of this study is the absence of control groups, such as those receiving only sensory-motor exercises, solely tDCS, or sham tDCS. The inclusion of these control groups would have enhanced our understanding of the results and their implications. Without them, our ability to accurately evaluate the real clinical effects of the interventions is limited. Specifically, it becomes challenging to assess the individual and combined effects of the interventions. This lack of comparative data obstructs our understanding of how effective each intervention is when analyzed independently.”

---

## [Decision Letter · Decision Letter 2]

11 Nov 2024

Effects of combining sensory-motor exercises with transcranial direct current stimulation on cortical processing and clinical symptoms in patients with lumbosacral radiculopathy: an exploratory randomized controlled trial

PONE-D-24-30889R2

Dear Dr. Khanmohammadi,

We’re pleased to inform you that your manuscript has been judged scientifically suitable for publication and will be formally accepted for publication once it meets all outstanding technical requirements.

Kind regards,

Abdolvahed Narmashiri

Academic Editor

PLOS ONE

Additional Editor Comments (optional):

Reviewers' comments:

Reviewer's Responses to Questions

**Comments to the Author**

1. If the authors have adequately addressed your comments raised in a previous round of review and you feel that this manuscript is now acceptable for publication, you may indicate that here to bypass the “Comments to the Author” section, enter your conflict of interest statement in the “Confidential to Editor” section, and submit your "Accept" recommendation.

Reviewer #1: All comments have been addressed

2. Is the manuscript technically sound, and do the data support the conclusions?

Reviewer #1: (No Response)

3. Has the statistical analysis been performed appropriately and rigorously? 

Reviewer #1: (No Response)

4. Have the authors made all data underlying the findings in their manuscript fully available?

Reviewer #1: (No Response)

5. Is the manuscript presented in an intelligible fashion and written in standard English?

Reviewer #1: (No Response)

6. Review Comments to the Author

Reviewer #1: (No Response)

7. PLOS authors have the option to publish the peer review history of their article (what does this mean?). If published, this will include your full peer review and any attached files.

Reviewer #1: No

---

## [Editor Report · Acceptance letter]

11 Dec 2024

PONE-D-24-30889R2 

PLOS ONE

Dear Dr. Khanmohammadi, 

I'm pleased to inform you that your manuscript has been deemed suitable for publication in PLOS ONE. Congratulations! Your manuscript is now being handed over to our production team.

Kind regards, 

on behalf of

Dr. Abdolvahed Narmashiri 

Academic Editor

PLOS ONE